# Socioeconomic differences in the impact of prices and taxes on tobacco use in low- and middle-income countries–A systematic review

G. Emmanuel Guindon[1,2,3]*, Umaima Abbas[1], Riya Trivedi[1], Sophiya Garasia[1,2], Sydney Johnson[1], Rijo M. John[4]

**1** Centre for Health Economics and Policy Analysis, McMaster University, Hamilton, Ontario, Canada, **2** Department of Health Research Methods, Evidence, and Impact, McMaster University, Hamilton, Ontario, Canada, **3** Department of Economics, McMaster University, Hamilton, Ontario, Canada, **4** Rajagiri College of Social Sciences, Rajagiri PO, Kochi, Kerala, India

* emmanuel.guindon@mcmaster.ca

**Data Availability Statement:** All data generated or analyzed during this study are included in this

## Abstract

There is indisputable evidence that increases in taxes that raise tobacco prices reduce tobacco use. Consumption taxes on manufactured tobacco products, however, can be regressive in socioeconomic status (e.g., when the ratio of tax paid to income is lower for higher-income groups than for lower-income groups). Nevertheless, if the poor or less educated are more price responsive, a change in tobacco tax may be progressive in socioeconomic status. Existing reviews clearly indicate that populations with lower income or education are more responsive to tobacco tax and price changes than higher-income and more educated populations in high-income countries. Research pertaining to low- and middle-income countries was, however, limited and inconclusive. We conducted a review of quantitative studies that examined if socioeconomic status modified the association between prices and taxes and tobacco use in low- and middle-income countries. We searched two electronic databases, two search engines, and two working paper repositories. At least two reviewers independently screened articles for inclusion, extracted detailed characteristics, and assessed the risk of bias of each included study. Thirty-two studies met our inclusion criteria. Overall, we found that the evidence in low- and middle-income countries was too limited and methodologically weak to make any conclusive statements. Our review highlights a number of data and methodological limitations in existing studies. The most important limitation was the lack of formal assessment of socioeconomic differences in price responsiveness. Only seven of 32 studies assessed statistically whether own-price effects were modified by socioeconomic status. Many modelling studies have examined the distributional effect of a tax increase on tobacco use, while assuming a strong own-price elasticity gradient in income. The poor were generally assumed to be more responsive to price by a factor of two to five, relative to the wealthy. Although there are theoretical reasons to expect poorer individuals to be more responsive to monetary prices than wealthy ones in low- and middle-income countries, our review provides little empirical support.

published article [and its supplementary information files].

**Funding:** Canadian Institutes for Health Research (grant # 149763). GEG holds the Centre for Health Economics and Policy Analysis (CHEPA)/Ontario Ministry of Health and Long-Term Care (MOHLTC) Chair in Health Equity, an endowed Chair funded in part by the MOHLTC. The funder had no role in the study design, analysis, interpretation, writing of the report, or in the decision to submit this article for publication.

**Competing interests:** The authors have declared that no competing interests exist.

## Introduction

It is estimated that nearly 6 million deaths were attributable to smoking tobacco use in low- and middle-income countries (LMICs) in 2019 [1]. Of all deaths attributable to smoking tobacco use, nearly 80% occurred in LMICs [1]. Associations between socioeconomic status (SES) and tobacco use are well documented in high-income countries [2]. Similar associations also exist in LMICs [3]. A recent systematic review that examined SES and non-communicable disease behavioural risk factors in low-income and lower-middle-income countries found that low socioeconomic groups had a significantly higher prevalence of tobacco use [4].

There is indisputable evidence that increases in taxes that raise tobacco prices reduce tobacco use in low-, middle-, and high-income countries [5–7]. Consumption taxes, such as those imposed on manufactured tobacco products (e.g., cigarettes) can be regressive in SES; the most common definitions of regressivity being based on tax burden or the 'ability to pay' such that the ratio of tax paid to income is lower for higher-SES groups than for lower-SES groups [8].

Although the poor inevitably have lower income and more of them tend to use tobacco, they may buy lower-priced tobacco products so that lower-income tobacco users may spend less on tobacco than higher-income tobacco users. Additionally, theory suggests the poor may be more responsive to changes in monetary price. First, the poor may spend a larger share of their relatively smaller income on tobacco than wealthier tobacco users. Second, lower-income individuals may behave differently when it comes to choices involving intertemporal tradeoffs. Economic models of addiction suggest that individuals with a greater preference for the present (lower-income and less educated) may be relatively more sensitive to changes in the monetary price of addictive goods [9]. As a result, if the poor are more sensitive to price changes, even if a tobacco tax is regressive, a change in tobacco tax may be progressive. Additionally, the overall monetary effect of a tax change may be positive for the poor once benefits accrued through lower medical expenses and an increase in working years are taken into consideration [10].

In its seminal 1999 report, the World Bank concluded there was evidence that the poor were more responsive to changes in tobacco prices in high-income countries but that further studies in LMICs were required to confirm this finding [11]. In 2011, the International Agency for Research on Cancer (IARC), after a compressive examination of the effectiveness of tobacco price and tax policies concluded there was 'limited evidence' that lower-income populations were more responsive to tobacco tax and price increases than higher-income populations in LMICs [5]. In its 2011 report, IARC identified 16 LMIC studies: two from Africa, three from eastern Asia, five from south-eastern Asia, three from southern Asia, one from western Asia, and three from eastern Europe. No studies were identified from Latin America and the Caribbean, Central Asia or Oceania (S1 Appendix) [5]. Of the 16 studies, eight provided some support that lower-income populations in LMICs may have been more responsive to changes in tobacco prices than higher-income populations [12–19]. However, none of the eight studies formally examined if SES differences in price responsiveness were statistically significant, and only two reported uncertainty intervals [14, 19]. Of the two studies that reported uncertainty intervals, one reported nonsensical positive own-price elasticities for higher-income individuals [19], while the other reported wide and overlapping 95% confidence intervals, so it is unlikely that any differences were statistically significant [14].

Recent reviews that examined the impact of price or tax on tobacco typically focused solely on high-income countries [20–22] or pointed to a dearth of studies that used data from LMICs [23]. A 2021 review of south-eastern Asian countries included only five studies that had examined socioeconomic differences in price responsive, and only one published after 2010 [24].

Similarly, a 2015 review of Latin America and the Caribbean countries identified a single study that had reported price effects by SES categories [6].

Policymakers' hesitancy to increase tobacco taxes, despite the overwhelming evidence that higher prices reduce tobacco use and improve health, is often due to the common argument that higher tobacco taxes may be regressive in SES [8]. A common flaw in assessing regressivity based solely on tax burden by SES groups is that it ignores the behavioural responses to taxation [8]. Consequently, a proper assessment of the benefits and costs of tobacco taxation requires knowledge about individuals respond when faced with higher taxes and prices. Our objective was to systematically review quantitative studies that examined if socioeconomic status modified the association between prices and taxes and tobacco use in LMICs.

## Methods

### Searches

We searched, from January 1, 2010, two electronic bibliographic databases (MEDLINE and EconLit), two search engines (Google and Google Scholar), and two working paper repositories (RePEc [Research Papers in Economics] and the National Bureau of Economic Research Working Paper series). The database searches were last updated on April 18, 2022. Additionally, we examined references of included studies and used Google Scholar to examine studies that referenced included studies. The search strategy is provided in the S2 Appendix. At least two reviewers independently screened the articles for inclusion, extracted detailed characteristics, and assessed the risk of bias of each included study. Conflicts were resolved by involving additional reviewers.

### Inclusion and exclusion criteria

We included all quantitative studies that examined the association between prices, taxes and tobacco use and reported socioeconomic differences in price responsiveness to tobacco use in LMICs. We included all measures of tobacco use (initiation or onset, participation, consumption, cessation, substitution, escalation, or persistence) and excluded aggregate outcome measures, such as national cigarette consumption or sales. We defined socioeconomic status broadly and included measures such as income, wealth, education, and occupation and excluded studies that solely examined differences between urban and rural areas. We included all measures of associations, including but not limited to: own-price elasticities, odds ratios, hazard ratios and marginal effects.

We used the World Bank Analytical Classifications to exclude high-income countries. As countries can move between categories over time, we used the classification for the year(s) in which data were collected. We excluded cross-country studies if it were not possible to disentangle results between LMICs and high-income countries. Given IARC's compressive review was published in 2011, we only included studies published from 2010. Table A1 in S1 Appendix presents detailed results for each LMIC study included in the IARC review.

### Data extraction, risk of bias assessment, and data synthesis

Studies that examine associations between prices, taxes and tobacco use often use methodological approaches that are overlooked in risk of bias assessment tools. Examples include demand system approaches such as the Almost Ideal demand system, instrumental variable and difference-in-differences approaches and, duration/survival analyses. We used appraisal elements proposed for social experiments and quasi-experiments, and borrowed critical appraisal criteria from the Maryland Scientific Methods Scale and the Cochrane Effective Practice and

Organisation of Care (EPOC) Risk of bias tools [25–28]. Due to heterogeneity between studies, a meta-analysis was not possible. Important differences between studies included outcomes (e.g., initiation, participation, consumption), population (e.g., youth, adults, smokers), measures of prices and taxes (e.g., contemporaneous/spacial differences in prices, temporal changes prices or taxes), and analytical approaches (e.g., demand systems using budget shares and unit values, instrumental variable estimation, survival analysis, two-part models).

The following study characteristics were extracted and summarized: citation, country, competing interests and funding disclosure, data type, method, outcomes, price/tax measure, and covariates, sensitivity or robustness checks, and detailed results for each outcome. We assessed the following elements, if applicable:

- clear reporting of the dependent variable(s), explanatory variables, adjustment for inflation; methods, and results (including measures of uncertainty); the source of funding and competing interests;

- clear reporting of the source and extent of price or tax variation (i.e., was there enough variation to identify the effect of price or tax on tobacco use);

- misspecification tests and/or sensitivity analyses that were conducted;

- the extent of missing data, outliers, and attrition and how they were handled;

- was measurement error, quality, or endogeneity taken into account;

- when duration analyses were used, was there any informative censoring, was the functional form of time/duration dependency clearly reported and appropriate, were prices matched to retrospective data appropriately, was the assumption that everyone eventually fails examined;

- was there a formal assessment of socioeconomic differences in price responsiveness;

- precision: how narrow/wide were uncertainty intervals;

- conclusions: were the study conclusions supported by results.

We did not contact authors of primary studies if information was missing or unclear. We did not use scores or scales for assessing the overall risk of bias, as empirical evidence does not support them [29]. For example, blindly adding distinct elements assumes that each element is worth the same, while assigning weights to each element disallows any variation in the bias of each element. Although unclear reporting may be a signal for poor execution, more weight was assigned to our assessment of the methodological approaches. We followed ROBINS-I (a tool for assessing risk of bias in non-randomised studies of interventions) and assessed studies as 'low risk of bias' (the study is comparable to a well-performed randomized trial), 'moderate risk of bias' (the study is sound for a non-randomized study but cannot be considered comparable to a well-performed randomized trial), 'serious risk of bias' (the study has some important problems), and 'critical risk of bias' (the study is too problematic to provide any useful evidence on the effects of intervention) [30].

To assess confidence in our outcome (i.e., socioeconomic differences in the impact of prices and taxes on tobacco use in LMICs), we considered precision of the effect estimate, consistency of findings across studies, and study design limitations (in particular if socioeconomic differences in price responsiveness were formally assessed using a statistical test). A review protocol was prepared in advance as a part of a funding proposal but was not publicly registered. No amendment were made to the review protocol. At least two reviewers independently extracted detailed characteristics, and assessed the risk of bias of each included study. We used

structured summaries to present results (due to heterogeneity between studies a meta-analysis was not possible). A completed PRISMA 2020 Checklist is available in S1 Checklist [31].

## Results

The database search produced 1160 records after the removal of duplicate citations, from which 1123 were excluded based on the title/abstract screen and 16 were subsequently removed after a full-text screen, yielding 32 studies that met all inclusion criteria (Fig 1). Key study characteristics and our risk of bias assessment are presented in Table 1. More detailed study characteristics of included studies are presented in the S3 Appendix. Out of the 32

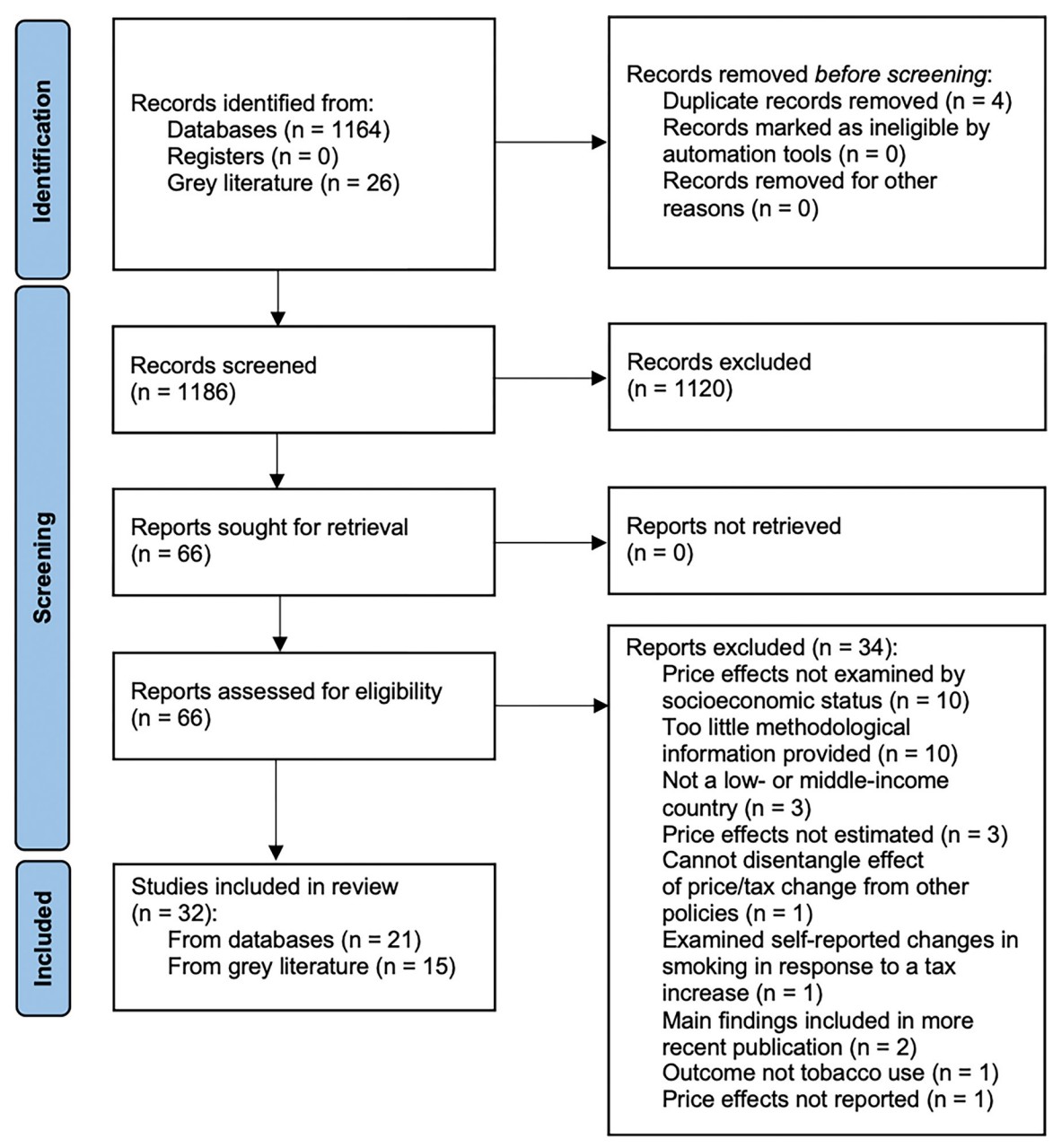

**Fig 1. Identification of studies.**

**Table 1. Characteristics and risk of bias of included studies.**

| Authors/year; journal country; competing interests; funding | Data type; method | Outcomes | Price/tax measure | Covariates | Risk of bias |
|---|---|---|---|---|---|
| **Africa: Northern Africa** (no. of studies: 0) | | | | | |
| **Africa: Sub-Saharan Africa** (no. of studies: 2) | | | | | |
| Dauchy, Ross, 2019; Addiction [33] –Kenya Competing interests: disclosed (none reported) Funding: IDRC | Retrospective constructed from cross-sectional data (2014) Pooled linear probability model (initiation: with propensity score matching) Functional form of duration dependency not clearly described; age, age squared included as covariates | Smoking initiation; no clear definition provided Age at first risk of starting: unclear Cessation: no clear definition provided | Real prices of Crown Bird (1960–1967) and Sportman (1968–2014) cigarettes Deflator: CPI all-items | Age; urban/rural; marital status; education; occupation; wealth. | • unclear description of dependent variables • no testing for misspecification • functional form of duration/ time dependency not clearly reported • unclear how prices were matched to retrospective individual-level data • imprecision: uncertainty intervals very wide • Risk of bias: serious |
| Kidane, Hepepwa et al., 2017; Applied Econometrics [32] –Tanzania Competing interests: not disclosed Funding: US National Institutes of Health-Fogarty International Center; US National Cancer Institute | Repeated cross-sectional (2008, 2010, 2012) Two-part model: Participation: logistic regression Consumption: OLS | Participation: currently smoking; 'smoking' not defined Consumption: packs of cigarettes smoked per month, conditional on smoking | Regional unit values in 2008 (total cigarette expenditure/sticks purchased); # of regions not reported; 2010, 2012 prices estimated by multiplying 2008 prices by CPI cigarette; converted to packs by taking the average quantity of three representative brands Deflator: unclear | Sex; residence; age, education; total annual expenditure. | • unclear description of dependent variable • unclear reporting of missing data/outliers • no. of clusters not reported • unclear adjustment for inflation • no testing for misspecification • no sensitivity analyses • no account for measurement error or quality • imprecision: unclear; likely very high (measures of uncertainty not reported) • no formal assessment of socioeconomic differences in price responsiveness • differences in own-price elasticities across time and between expenditure groups not credible • conclusion not supported by results. • Risk of bias: critical |
| **Americas: Caribbean** (no. of studies: 0) | | | | | |
| **Americas: Latin America** (no. of studies: 11) | | | | | |
| Guindon, Paraje, Chávez, 2018; Econ Inq [38] –Argentina Competing interests: disclosed (none reported) Funding: IDRC | Retrospective constructed from cross-sectional data (2005, 2008, 2009, 2011) Survival/duration analyses (discrete-time hazard models and a complementary loglog specification) Functional form of duration dependency: cubic polynomial | Age at smoking onset Age at first risk of starting: 8 | Manufactured tobacco component of the CPI for Greater Buenos Aires (Jan 1980—May 2008); after-tax monthly weighted average price for a pack of 20 cigarettes (> May 2008) Deflator: CPI all-items Greater Buenos Aires (1980–2006); CPI all-items, Sante Fe province (> 2006); MIT's Billion Prices Project (> 2006) | Alcohol prices; periods of hyperinflation and very high inflation; sex; household head's education level; 1986 national tobacco control policies; province-level smoke free policies; provinces; survey wave. | • no testing for misspecification • informative censoring among younger survey respondents • imprecision: uncertainty intervals fairly wide • Risk of bias: moderate |

*(Continued)*

**Table 1.** (Continued)

| Authors/year; journal country; competing interests; funding | Data type; method | Outcomes | Price/tax measure | Covariates | Risk of bias |
|---|---|---|---|---|---|
| Divino, Ehrl et al., 2021; Tob Control [41] –Brazil Competing interests: disclosed (none reported) Funding: Bloomberg Philanthropies; University of Illinois Chicago | Repeated cross-sectional data (2008, 2013) Two-part model: Participation: Probit; Consumption: unclear | Participation: smoker or not; 'smoker' not defined Consumption: no. of cigarettes smoked per day, conditional on smoking | Mean of self-reported prices across smokers at federal state level; no. of states not reported; Brazil has 26 states (estados) and one federal district (distrito federal) Deflator: unclear | Income; age; education; years of smoking; sex; federal state. | • unclear description of dependent variables, data and methods • unclear reporting of missing data/outliers • unclear adjustment for inflation • no. of clusters not clearly reported; small no. of clusters ($\leq$ 27) and time periods (2) • no testing for misspecification • no sensitivity analysis • imprecision: uncertainty intervals fairly wide • no formal assessment of socioeconomic differences in price responsiveness • Risk of bias: critical |
| Guindon, Paraje, Chaloupka, 2019; JAMA Pediatr [39] –Chile Competing interests: disclosed (none reported). Funding: IDRC | Retrospective constructed from cross-sectional data (2001, 2003, 2005, 2007, 2009, 2011, 2013, 2015) Survival/duration analyses (discrete-time hazard models and a complementary loglog specification) Functional form of duration dependency: cubic polynomial | Age at smoking onset Age at first risk of starting: 8 | Cigarette component of the CPI Deflator: CPI all-items | Sex; mother's educational level; type of school; 2006 tobacco control law; regions; alcohol prices. | • no testing for misspecification • informative censoring • Risk of bias: moderate |
| Gallego, Otalvaro-Ramirez, Rodriguez-Lesmes, 2021; Tob Control [43] –Colombia Competing interests: disclosed (none reported). Funding: IDRC | Repeated cross-sectional data (2008, 2013) Logistic regression | Participation (unclear how it was defined) | State-level (9 departamentos) prices constructed from household-level unit values (2006–2007) and CPI for tobacco/ cigarettes Deflator: unclear | Household head alcohol consumption, ever tried marijuana, sex, marital status, employment status, education, SES; year-month; municipality. | • dependent variable not clearly defined • unclear adjustment for inflation • unclear how missing data were handled • no testing for misspecification • limited variation in space (9 states) and time (2 survey cycles) • imprecision: uncertainty intervals very wide • no formal assessment of socioeconomic differences in price responsiveness • Risk of bias: critical |

(*Continued*)

**Table 1.** (Continued)

| Authors/year; journal country; competing interests; funding | Data type; method | Outcomes | Price/tax measure | Covariates | Risk of bias |
|---|---|---|---|---|---|
| Chávez, 2016; Rev Panam Salud Publica [36] –Ecuador Competing interests: disclosed (none reported) Funding: Escuela de Negocios de la Universidad Adolfo Ibáñez, American Cancer Society; IDRC | Cross-sectional (Apr 2011—Mar 2012); Deaton's two-equation system of budget shares and unit values | Share of the budget devoted to: cigarettes alcohol (not clearly defined) | Unit values (expenditures/quantity consumed) at cluster-level (624 *parroquias* [parishes]) Deflator: unclear | Not clearly presented. | • no testing for misspecification • covariates not clearly presented • unclear how missing data/outliers were handled • unclear adjustment for inflation • no. of household with positive cigarette/alcohol expenditures within each cluster not clearly reported; no. of clusters with at least two household with positive cigarette/alcohol expenditures not clearly reported • imprecision: uncertainty intervals very wide • no sensitivity analyses • no formal assessment of socioeconomic differences in price responsiveness • Risk of bias: serious |
| Paraje, Araya et al., 2021; Tob Control [44] –El Salvador Competing interests: disclosed (none reported) Funding: International Union Against Tuberculosis and Lung Disease; Pan-American Health Organization | Cross-sectional (2005/2006) Almost Ideal Demand System (AIDS) | Share of the budget devoted to cigarettes | Unit values (expenditures/quantity consumed) at cluster-level (468 primary sampling units) Deflator: unclear | No. of individuals ≥ 15 years in household; share of women ≥ 15 years in household; household head education, sex, and age. | • unclear adjustment for inflation • no. of clusters not reported; no. of household with positive cigarette expenditures within each cluster not clearly reported • no account for measurement error • no testing for misspecification • no sensitivity analysis • imprecision: uncertainty intervals very wide • no formal assessment of socioeconomic differences in price responsiveness • Risk of bias: serious |
| Franco-Churruarin, Gonzalez-Rozada, 2021; Report (University of Illinois Chicago) [42] –Mexico Competing interests: not disclosed Funding: Bloomberg Philanthropies | Cross-sectional (2015) Two-part model: Participation: probit (prices in ln) | Participation: daily smokers (individuals who self-reported smoking and who smoked a positive number of cigarettes each day) | 1) Self-reported price paid for the last purchase assigned to smokers; random regression imputation (stochastic regression imputation) to assign price to non-smokers 2) Mean self-reported price by primary sampling unit (PSU) assigned to smokers and non-smokers (no. of PSUs not reported) Deflator: n/a | Wealth index; sex; age; rural/urban; student; employment (employed, unemployed, out of labour force). | • unclear how missing data were handled • no. of clusters not reported • imprecision: uncertainty intervals fairly wide • no formal assessment of socioeconomic differences in price responsiveness • conclusion not supported by results • Risk of bias: serious |

(*Continued*)

**Table 1.** (Continued)

| Authors/year; journal country; competing interests; funding | Data type; method | Outcomes | Price/tax measure | Covariates | Risk of bias |
|---|---|---|---|---|---|
| Sáenz de Miera Juárez, Guerrero López et al., 2013; Report (Fundación InterAmericana del Corazón México) [35] –Mexico Competing interests: not disclosed Funding: IDRC | Repeated cross-sectional (1994, 1996, 1998, 2000, 2002, 2004, 2005, 2006, 2008, 2010, 2012) Two-part model: Participation: probit Consumption: weighted-OLS | Participation: positive cigarette expenditures Consumption: monthly cigarette consumption (calculated from weekly cigarette consumption measured in kg using conversion factor 1 kg = 800 cigarettes) | Predicted unit values (expenditures/quantity consumed) using categorical indicators of household income (quintiles), rural/urban status; state Deflator: unclear | Income; household head's education, sex, age; alcohol use; no. of adults in household; survey wave. | • unclear adjustment for inflation • no account for measurement error or endogeneity • no testing for misspecification • no sensitivity analyses • imprecision: uncertainty intervals fairly wide; not all uncertainty intervals reported • no formal assessment of socioeconomic differences in price responsiveness • Risk of bias: serious |
| Saenz-de-Miera, Thrasher et al., 2010; Tob Control [34] –Mexico Competing interests: disclosed (none reported) Funding: Consejo Nacional de Ciencia y Tecnología Mexico; US National Institute of Health | Longitudinal (wave 1: Sep-Nov 2006, wave 2: Nov-Dec 2007) t-tests (average no. of cigarettes smoked per day at baseline and follow-up) Logistic regressions (quitting at follow-up) | Average no. of cigarettes smoked per day at baseline and follow-up Quitting at follow-up | Pre-post tax changes: tax increased from 110% of the price to the retailer to 140%; magnitude of price change is unclear (average self-reported cigarette pack price increased among smokers whose last purchase was a pack at both survey waves was 12.7% (inflation adjusted) Deflator: unclear | Average no. of cigarettes smoked per day at baseline and follow-up: unclear, likely not adjusted. Quitting at follow-up: age; sex; marital status; education; employment status; income; smoking intensity; quit attempts in past year; plan to quit in next six months. | • unclear how missing data were handled • non-random attrition • unclear adjustment for inflation • unclear how change in taxes affected prices • no testing for misspecification; • imprecision: uncertainty intervals very wide; not all uncertainty intervals reported • no formal assessment of socioeconomic differences in price responsiveness • limited generalizability of findings (respondents selected for 4 urban centres) • Risk of bias: serious |
| de los Ríos, Medina, Aguilar, 2020; Working Paper (Instituto de Estudios Peruanos) [40] –Peru Competing interests: not disclosed Funding: not disclosed | Cross-sectional (2009) Two-part model: Participation: probit Consumption: Deaton's two-equation system of budget shares and unit values | Participation: positive cigarette expenditures Consumption: share of the budget devoted to cigarettes | Unit values (expenditures/quantity consumed) at cluster-level (249 districts) Deflator: unclear | Household share of/with: women/men; higher education/secondary education; highest level of education reached by any member; working age; hh head education, sex, age, working status. | • unclear description of dependent variables and covariates • unclear reporting of missing data/outliers • unclear adjustment for inflation • no. of household with positive cigarette expenditures within each cluster not clearly reported; clusters (districts) with only 1 household with positive cigarette expenditures included in consumption part; clusters with 2 or more included in both participation and consumption parts • no testing for misspecification • no sensitivity analysis • imprecision: uncertainty intervals very wide • no formal assessment of socioeconomic differences in price responsiveness. • Risk of bias: serious |

(*Continued*)

**Table 1.** (Continued)

| Authors/year; journal country; competing interests; funding | Data type; method | Outcomes | Price/tax measure | Covariates | Risk of bias |
|---|---|---|---|---|---|
| Gonzalez-Rozada, Ramos-Carbajales, 2016; Rev Panam Salud Publica [37] –Peru Competing interests: disclosed (none reported) Funding: IDRC | Cross-sectional (2008–2009) Deaton's two-equation system of budget shares and unit values | Share of the budget devoted to cigarettes | Unit values (expenditures/quantity consumed) at cluster-level (clusters not defined; no. of clusters not reported) Deflator: unclear | No. of persons in household; % of men and women > 18 years in household. | • unclear how missing data/outliers were handled • unclear adjustment for inflation • clusters not defined; no. of clusters not reported • no testing for misspecification • imprecision: uncertainty intervals very wide • no formal assessment of socioeconomic differences in price responsiveness • Risk of bias: serious |
| **Asia: Central Asia** (no. of study: 0) | | | | | |
| **Asia: Eastern Asia** (no. of study: 2) | | | | | |
| Huang, Zheng et al., 2015; Tob Control [46] –China Competing interests: disclosed (none reported) Funding: US National Cancer Institute, Roswell Park Transdisciplinary Tobacco Use Research Centre, Robert Wood Johnson Foundation, CIHR, Chinese Centre for Disease Control and Prevention | Repeated cross-sectional (2009; 2015) Multivariable linear analyses using generalized estimating equations | Average no. of cigarettes consumed per day | Self-reported price-per-pack, aggregated at cluster level (city district level) Deflator: prices adjusted for inflation, but deflator not reported | Sex; education; employment; income; age; marital status; interview years; city. | • unclear reporting of missing data/outliers • no. of clusters not reported • no testing for misspecification • no sensitivity analyses • Risk of bias: serious |
| Chen, Xing, 2011; China Economic Review [48] –China Competing interests: not disclosed Funding: World Health Organization, Bloomberg Philanthropies; Johns Hopkins University | Repeated cross-sectional (1999–2001) Deaton's two-equation system of budget shares and unit values. | Share of the budget devoted to cigarettes | Unit values (expenditures/quantity consumed) at cluster-level (24 clusters; 8 provinces, 3 years) Deflator: unclear | Not clearly described. | • unclear reporting of covariates • unclear how missing data/outliers were handled • very low no. of cluster • no testing for misspecification • no sensitivity analyses; • measures of uncertainty/significance level not reported • no formal assessment of socioeconomic differences in price responsiveness • Risk of bias: critical |
| **Asia: South-eastern Asia** (no. of studies: 3) | | | | | |
| Cheng, Estrada, 2020; Prev Med [45] –Philippines Competing interests: disclosed (none reported) Funding: US National Cancer Institute, Roswell Park Transdisciplinary Tobacco Use Research Centre; Robert Wood Johnson Foundation; CIHR; Chinese Centre for Disease Control and Prevention. | Repeated cross-sectional (2009, 2015) Two-part model: Participation: probit; IV-probit Consumption: OLS; two-stage least squares IV: weighted tax-per-stick computed using the volume of removals reported by Philippine Bureau of Internal Revenue (price observations that were equivalent or higher than the computed weighted tax-per-stick were dropped) | Participation: unclear ('smoked daily and less than daily') Consumption: 'number of sticks smoked daily' | Average price-per-stick (quantity in pack/price of recent purchase) of each respondent's primary sampling unit or type of residence (if primary sampling unit not available) Deflator: unclear | Sex; education; employment status; wealth; age; urban/rural; exposure to media relating to the dangers of smoking cigarettes; addiction. | • unclear reporting of missing data/outliers • no. of clusters not reported • no testing for misspecification • measures of uncertainty not reported • results extremely sensitive to alternative specifications • nonsensical positive conditional own-price elasticities • no formal assessment of socioeconomic differences in price responsiveness • Risk of bias: critical |

*(Continued)*

**Table 1.** (Continued)

| Authors/year; journal country; competing interests; funding | Data type; method | Outcomes | Price/tax measure | Covariates | Risk of bias |
|---|---|---|---|---|---|
| Quimbo, Casorla et al., 2012; Report (International Union Against Tuberculosis and Lung Disease) [47] –Philippines Competing interests: not disclosed Funding: Bloomberg Philanthropies; Bill and Melinda Gates Foundation | Cross-sectional (2003) Two-Stage Least Squares (instrument: regions [no description or no. of regions provided]) | Mean household cigarette consumption, per household member (consumption obtained from cigarette expenditures / average price at province-level) | Average cigarette prices in 2003 at province-level (no. of provinces not clearly reported; likely 17 or 56 provinces) Deflator: n/a | Total household expenditures; household head's age, sex, employment status, and education; expenditures on insurance (proxy for risk taking). | • unclear how missing data/outliers were handled <br>• no. of clusters (provinces) not clearly reported <br>• no account for quality <br>• limited variation in space (17 or 57 provinces) and time (no variation in time) <br>• unclear if instrument was valid <br>• instrument (regions) likely correlated with dependent variable <br>• imprecision: uncertainty intervals not reported <br>• no formal assessment of socioeconomic differences in price responsiveness <br>• Risk of bias: critical |
| Jankhotkaew, Pitayarangsarit et al., 2021; Tob Control [49] –Thailand Competing interests: disclosed (none reported) Funding: disclosed (none reported) | Cross-sectional (2017) Two-part model: Participation: probit (prices in ln) Consumption: OLS (consumption/prices in ln) | Participation: no clear definition provided; Consumption: "number of cigarette sticks bought in the last purchase" | Cigarettes/roll-your-own: unit values at cluster level (village) Deflator: n/a | Income; sex; age, highest educational achievement; marital status; employment status; rural/urban; region. | • unclear description of dependent variables <br>• 18% of eligible respondent excluded because of missing data <br>• clusters not clearly defined; no. of clusters not reported <br>• no account for measurement error or quality <br>• no testing for misspecification <br>• nonsensical positive conditional own-price elasticities <br>• imprecision: measures of uncertainty not reported <br>• no formal assessment of socioeconomic differences in price responsiveness <br>• Risk of bias: critical |
| **Asia: Southern Asia** (no. of studies: 8) | | | | | |
| Huque, Abdullah et al., 2021; Report (ARK Foundation) [55] –Bangladesh Competing interests: disclosed (none reported) Funding: Bloomberg Philanthropies; University of Illinois Chicago | Repeated cross-sectional (2009, 2017) Two-part model: Participation: probit (prices in ln) Consumption: unclear | Participation: no clear definition provided; "people who smoke cigarettes daily or less than daily" Consumption: no clear definition provided; "number of sticks smoked" | Unit values at cluster level (weighted by individual cigarette expenditure) Deflator: n/a | Biri unit values at cluster-level; wealth index; sex; age; rural/urban; education; employment type; age; "smoking restrictions in homes (self-imposed) and workplaces (legislation or regulation imposed by authority), exposure to smoking warnings, advertising, promotional activities, and perceptions;" unclear how smoking restrictions, exposure to warnings/advertising were operationalized. | • covariates not clearly described <br>• clusters not clearly defined; no. of clusters not reported <br>• no account for measurement error or quality <br>• no testing for misspecification <br>• imprecision: uncertainty intervals fairly wide <br>• no formal assessment of socioeconomic differences in price responsiveness <br>• Risk of bias: critical |

(*Continued*)

**Table 1.** (*Continued*)

| Authors/year; journal country; competing interests; funding | Data type; method | Outcomes | Price/tax measure | Covariates | Risk of bias |
|---|---|---|---|---|---|
| Del Carmen, Fuchs, Genoni, 2018; Working Paper (World Bank) [53] –Bangladesh Competing interests: not disclosed Funding: World Bank; Bill and Melinda Gates Foundation; Bloomberg Philanthropies. | Cross-sectional (2016–2017) Quadratic Almost Ideal Demand System (QUAIDS) | Share of the budget devoted to: cigarettes bidis betel leaf betel nut rolled betel leaf | Median unit values (monthly expenditure/ quantity purchased) at district level in urban and rural areas. In cases where the no. of observations was less than 30, the medians of the full districts were used. Deflator: n/a | Household size; no. of males (≥ 15 years) in household; age, sex, religion, educational attainment of household head; urban/rural. | • unclear how missing data/ outliers were handled • no. of household with positive cigarette/bidi expenditures within each cluster not clearly reported; no. of clusters with at least two household with positive cigarette/bidi expenditures not clearly reported • no account for measurement error or endogeneity • no testing for misspecification • no sensitivity analyses • imprecision: uncertainty intervals not reported • no formal assessment of socioeconomic differences in price responsiveness • Risk of bias: critical |
| Nargis, Ruthbah, et al., 2014; Tob Control [51] –Bangladesh Competing interests: disclosed (none reported) Funding: IDRC; CIHR; US National Cancer Institute; Roswell Park Transdisciplinary Tobacco Use Research Center; Robert Wood Johnson Foundation; Ontario Institute for Cancer Research | Repeated cross-sectional (2009, 2010) Two-part model: Participation: probit; IV-probit Consumption: weighted-OLS; two-stage least squares | Participation (unclear how it was defined) Consumption, average number of cigarettes smoked per day | Prices self-reported by smokers, averaged at 'geographic area of residence (village)' Deflator: unclear | Household income; household size; sex; age; marital status; household size; education; occupation; household restriction on indoor smoking; survey wave; urban/rural. | • dependent variable not clearly defined • unclear how missing data/ outliers were handled • unclear adjustment for inflation • clusters not clearly defined; no. of clusters not reported; no. of self-reported prices per cluster not reported • imprecision: uncertainty intervals not clearly reported • socioeconomic differences in price responsiveness not formally assessed • conclusion not supported by results • Risk of bias: critical |
| Dauchy, John, 2022; Prev Sci [57] –India Competing interests: disclosed (none reported) Funding: Campaign for Tobacco-free Kids | Retrospective constructed from cross-sectional data (2016–17) Pooled linear probability model (initiation: with propensity score matching) Functional form of duration dependency not clearly described; age, age squared included as covariates | Smoking initiation: unclear how it was operationalized Age at first risk of starting: unclear Cessation: how long has it been since you stopped smoking? | Annual average wholesale price indices of cigarettes/bidis (weighted average), and smokeless tobacco from 1980 Deflator: wholesale price indices all-goods | Wealth, age; sex; education level; caste; religion; marital status. | • unclear description of dependent variables • unclear how missing data were handled • no testing for misspecification • functional form of duration/ time dependency not clearly reported • unclear how prices were matched to retrospective individual level data; interval to match price data to survey data likely very wide • survey data used did not allow to separate bidi and cigarette smoking • weighted wholesale price data used as a proxy for retail prices; no price variation in space (a single annual price for whole of India) • initiation models assumed that everyone eventually failed • Risk of bias: critical |

(*Continued*)

**Table 1.** (Continued)

| Authors/year; journal country; competing interests; funding | Data type; method | Outcomes | Price/tax measure | Covariates | Risk of bias |
|---|---|---|---|---|---|
| Selvaraj, Srivastava, Karan, 2015; BMJ Open [52] –India Competing interests: disclosed (none reported) Funding: IDRC | Cross-sectional (2011–2012) Deaton's two-equation system of budget shares and unit values | Share of the budget devoted to: cigarettes bidis loose leaf tobacco | Unit values (expenditures/quantity consumed) at cluster-level Deflator: unclear | Household expenditure; household size; ratio of males in household ($\geq 15$ years), household head education, religion, social group (caste); urban/rural. | • unclear how missing data/outliers were handled • unclear adjustment for inflation • no testing for misspecification • no sensitivity analyses • no. of clusters not reported • imprecision: reported standard errors too small to be credible • no formal assessment of socioeconomic differences in price responsiveness • Risk of bias: critical |
| Guindon, Nandi, et al., 2011; Working Paper (NBER) [50] –India Competing interests: not disclosed Funding: Bill & Melinda Gates Foundation | Repeated cross-sectional (1993/94; 1999/00; 2000/01; 2001/02; 2003; 2004; 2004/05–2007/08) Multilevel regressions (share-log functional form) | Share of the budget devoted to: cigarettes bidis country liquor | Unit values (expenditures/quantity consumed) at cluster-level (districts) Deflator: Consumer Price Index for Industrial Workers | Urban/rural; share of men in household; share of adults in household; household head education, sex, age, religion; survey wave. | • no testing for misspecification • unclear how missing data were handled • no account for measurement error or quality • no formal assessment of socioeconomic differences in price responsiveness • Risk of bias: serious |
| Raei, Emamgholipour et al., 2021; Health Econ Rev [56] –Iran Competing interests: disclosed (none reported) Funding: Tehran University of Medical Sciences; Health Information Management Research Center | Repeated cross-sectional (2002–2017) Two-part model: Participation: probit Consumption: OLS | Participation, positive cigarette expenditures Consumption, unclear how it was defined | Unclear, likely unit values Deflator: unclear | Household total expenditures; household share of members who were: aged 15–18 and $\geq 65$, female aged 19 to 64; jobless, with at least primary education, university education; household head age and sex; divorce rate, unemployment rate at province-level; survey years. | • dependent variable not clearly defined • unclear how missing data/outliers were handled • no description of price variable; likely unit values (clusters not defined; no. of clusters not reported) • unclear adjustment for inflation • no testing for misspecification • no sensitivity analyses • no account for measurement error or quality (if unit values were used as proxy for prices) • socioeconomic differences in price responsiveness not formally assessed • Risk of bias: critical |
| Nayab, Nasir et al., 2020; Tob Control [54] –Pakistan Competing interests: disclosed (none reported) Funding: Bloomberg Philanthropies; University of Illinois Chicago | Cross-sectional (2015–16) Deaton's two-equation system of budget shares and unit values | Share of the budget devoted to: cigarettes chewed tobacco (composite commodity including saunf, naswar, gutka) | Unit values (expenditures/quantity consumed) at cluster-level (enumeration blocks/villages) Deflator: unclear | Household total expenditures; household size; mean household education; highest degree obtained by a member of the household; education of the head of the household; share of adults in the household; share of male members in the household; no. of earners in the household; region and province of residence. | • unclear how missing data/outliers were handled • unclear adjustment for inflation • no. of unit values per cluster not reported; no. of clusters with positive cigarette/chewing tobacco expenditures not reported • no testing for misspecification • no sensitivity analyses • imprecision: uncertainty intervals very wide • socioeconomic differences in price responsiveness not formally assessed • Risk of bias: serious |
| **Europe: Eastern Europe** (no. of studies: 6) | | | | | |

*(Continued)*

**Table 1.** (*Continued*)

| Authors/year; journal country; competing interests; funding | Data type; method | Outcomes | Price/tax measure | Covariates | Risk of bias |
|---|---|---|---|---|---|
| Gjika, Zhllima, Imami, 2019; Report (University of Illinois Chicago) [58] –Albania Competing interests: not disclosed Funding: Bloomberg Philanthropies; University of Illinois Chicago | Repeated cross-sectional (2014–2017) Two-part model: Participation: logit Consumption: Deaton's two-equation system of budget shares and unit values, GLM | Participation, positive expenditures on cigarettes vs. no expenditures on cigarettes Consumption, share of the budget devoted to cigarettes | Unit values (expenditures/quantity consumed) at cluster-level Deflator: unclear | Not clearly reported; total expenditures; "share of men and adults in the household, maximum or mean level of education and activity of the household members), region and settlement fixed effects and variables representing institutional changes relevant to cigarette consumption." | • covariates not clearly described • unclear how missing data/ outliers were handled • unclear adjustment for inflation • clusters not defined; no. of clusters not reported • unit values treated as market prices in participation component of two-part model; unclear why Deaton's two-equation system was not used to obtain total own-price elasticities; results using Deaton's approach not presented for price elasticity estimates by SES categories • no testing for misspecification • imprecision: uncertainty intervals very wide • selective reporting of results • no formal assessment of socioeconomic differences in price responsiveness. • Risk of bias: critical |
| Gligorić, Kulovac et al., 2022; Tob Control [63] –Bosnia and Herzegovina Competing interests: disclosed (none reported) Funding: Bloomberg Philanthropies; University of Illinois Chicago | Repeated cross-sectional (2007, 2011, 2015) Two-part model: Participation: logit Consumption: Deaton's two-equation system of budget shares and unit values, GLM | Participation, positive household monthly expenditures on cigarettes vs. no expenditures on cigarettes Consumption, share of the budget devoted to cigarettes | Unit values (expenditures/quantity consumed) at cluster-level Deflator: CPI | Total monthly expenditures; household size; age; sex; mean and highest education level of household members; household adult and male ratio; urban/rural; household type (employed, self-employed, pensioner, unemployed). | • unclear how missing data were handled • no sensitivity analyses • unit values treated as market prices in participation component of two-part model; unclear why Deaton's two-equation system was not used to obtain total own-price elasticities • Risk of bias: moderate |
| Prekazi, Pula, 2019; Report (University of Illinois Chicago) [60] –Kosovo Competing interests: not disclosed Funding: Bloomberg Philanthropies; University of Illinois Chicago | Repeated cross-sectional (2007–2017) Two-part model: Participation: logit Consumption: Deaton's two-equation system of budget shares and unit values, GLM | Participation, positive household monthly expenditures on cigarettes vs. no expenditures on cigarettes Consumption, share of the budget devoted to cigarettes | Unit values (expenditures/quantity consumed) at cluster-level Deflator: unclear | Not clearly reported; total expenditures; "share of men and adults in the household, maximum or mean level of education and activity of the household members), region and settlement fixed effects and variables representing institutional changes relevant to cigarette consumption." | • covariates not clearly described • unclear how missing data/ outliers were handled • unclear adjustment for inflation • unit values treated as market prices in participation component of two-part model; unclear why Deaton's two-equation system was not used to obtain total own-price elasticities; results using Deaton's approach not presented for price elasticity estimates by SES categories • no testing for misspecification; • imprecision: uncertainty intervals very wide • no formal assessment of socioeconomic differences in price responsiveness • Risk of bias: critical |

(*Continued*)

**Table 1.** (Continued)

| Authors/year; journal country; competing interests; funding | Data type; method | Outcomes | Price/tax measure | Covariates | Risk of bias |
|---|---|---|---|---|---|
| Cizmovic, Mugosa et al., 2022; Tob Control [62] –Montenegro Competing interests: disclosed (none reported) Funding: Bloomberg Philanthropies; University of Illinois Chicago | Repeated cross-sectional (2006–2015, 2017) Two-part model: Participation: logit Consumption: Deaton's two-equation system of budget shares and unit values, GLM | Participation, positive household monthly expenditures on cigarettes vs. no expenditures on cigarettes Consumption, share of the budget devoted to cigarettes | Unit values (expenditures/quantity consumed) at cluster-level Deflator: CPI | Total monthly expenditures; household size; adult and male ratio in household; highest education level among household members; household type (employed, pensioner, unemployed). | • unclear how missing data were handled • unit values treated as market prices in participation component of two-part model; unclear why Deaton's two-equation system was not used to obtain total own-price elasticities • relatively few clusters per survey/year ($\approx$ 15) • Risk of bias: serious |
| Najdova, 2019; Report (University of Illinois Chicago) [59] –North Macedonia Competing interests: not disclosed Funding: Bloomberg Philanthropies; University of Illinois Chicago | Repeated cross-sectional (2015–2017) Two-part model: Participation: logit Consumption: Deaton's two-equation system of budget shares and unit values, GLM | Participation, positive household monthly expenditures on cigarettes vs. no expenditures on cigarettes Consumption, share of the budget devoted to cigarettes | Unit values (expenditures/quantity consumed) at cluster-level Deflator: unclear | Not clearly reported; total expenditures; "share of men and adults in the household, maximum or mean level of education and activity of the household members), region and settlement fixed effects and variables representing institutional changes relevant to cigarette consumption." | • covariates not clearly described • unclear how missing data/ outliers were handled • unclear adjustment for inflation • unit values treated as market prices in participation component of two-part model; unclear why Deaton's two-equation system was not used to obtain total own-price elasticities; no. of clusters not reported • no testing for misspecification • imprecision: uncertainty intervals very wide • no formal assessment of socioeconomic differences in price responsiveness • Risk of bias: critical |
| Vladisavljević, Đukić et al., 2019; Report (University of Illinois Chicago) [61] –Serbia Competing interests: not disclosed Funding: Bloomberg Philanthropies; University of Illinois Chicago | Repeated cross-sectional (2006–2017) Two-part model: Participation: logit Consumption: Deaton's two-equation system of budget shares and unit values, GLM | Participation, positive household monthly expenditures on cigarettes vs. no expenditures on cigarettes Consumption, share of the budget devoted to cigarettes | Unit values (expenditures/quantity consumed) at cluster-level Deflator: CPI | Total expenditures; household size; urban/ rural; age; sex composition of the household; mean and maximum level of education of the household members; household type (employed, self-employed, pensioner, unemployed). | • unclear how missing data/ outliers were handled • unit values treated as market prices in participation component of two-part model; unclear why Deaton's two-equation system was not used to obtain total own-price elasticities • no. of clusters not clearly reported; • no testing for misspecification • imprecision: uncertainty intervals fairly wide • no formal assessment of socioeconomic differences in price responsiveness • Risk of bias: serious |

**Oceania**: (no. of studies: 0)

Note: CIHR, Canadian Institutes for Health Research; CPI, Consumer Price Index; GLM, generalized linear model; IDRC, International Development Research Center; IV, instrumental variable; NBER, National Bureau of Economic Research; OLS, ordinary least square. Geographical regions are based on continental regions, which are further subdivided into sub-regions (United Nations Statistics Division, https://unstats.un.org/unsd/methodology/m49/

studies, two were conducted using data from Sub-Saharan Africa (Kenya, Tanzania) [32, 33], 11 from Latin America (Argentina, Brazil, Chile, Colombia, Ecuador, El Salvador, Mexico, Peru) [34–44], two from eastern Asia (China) [45, 46], three from south-eastern Asia (Philippines, Thailand) [47–49], eight from Southern Asia (Bangladesh, India, Iran, Pakistan) [50–57], and six from eastern Europe (Albania, Bosnia and Herzegovina, Kosovo, Montenegro, North Macedonia, Serbia) [58–63]. We did not identify any studies from northern Africa, the Caribbean, central Asia, or Oceania.

## Study characteristics

Table 1 presents the characteristics and risk of bias of included studies. Studies are presented by geographic regions, and countries (in alphabetic order), in reverse chronological order (by date of publication). Most studies used cross-sectional or repeated cross-sectional data, while four studies used retrospective data constructed from cross-sectional data. Only one study used longitudinal data. Fifteen studies estimated two-part models, while three studies estimated one of the two parts (i.e., participation or consumption).

Five studies used demand system approaches; four used Deaton's two-equation system of budget shares and unit values that corrects for quality and measurement error, while two studies used the Almost Ideal Demand System (AIDS) of Deaton and Muellbauer. A further seven studies used Deaton's two-equation system of budget shares and unit values, but only for the consumption component of the two-part model. Four studies used duration analyses to examine the association between cigarette prices and smoking onset and/or cessation.

Most studies used unit values (expenditures/quantity consumed) or self-reported cigarette prices to construct a measure of price that varied in space. Four studies used time series of cigarette prices used in the construction of consumer or wholesale price indices. Only one study examined a change in tax.

Most studies clearly disclosed competing interests and funding; 11 studies clearly disclosed funding but not competing interests, while one study did not report competing interests or funding. Thirteen studies reported funding by the Bloomberg Philanthropies, a New York City based philanthropic organization and nine reported funding by the International Development Research Center, a Canadian Crown corporation that funds research in LMICs. None of the studies reported funding from tobacco manufacturers or any of their affiliates.

## Study findings

Key results and a summary of findings are presented in Table 2. Of 32 studies, only seven studies formally examined socioeconomic differences in price responsiveness and three found that lower-SES individuals or households were more responsive to cigarette prices than those of higher-SES. One study, rated at moderate risk of bias, found that households in Bosnia and Herzegovina with lower total expenditures were more responsive to cigarette prices, with differences large enough to be economically meaningful [63]. Another study, rated at serious risk of bias, found that low- and middle-SES households were more responsive to price than high-SES households in Montenegro [62]. A third study, rated at critical risk of bias, found some statistically significant socioeconomic differences in price responsiveness for cigarette/bidi smoking initiation in India; differences, however, were very small and unlikely to be meaningful [57].

Own-price elasticities estimates for a further eight studies suggest a potential gradient in SES. Studies rated at serious or critical risk of bias from Albania, Bangladesh, India, North Macedonia, Peru, the Philippines, and Serbia found that participation and/or consumption own-price elasticities tended to be higher for lower-SES households [40, 47, 49, 52, 55, 58, 59,

**Table 2. Summary of key findings.**

| Authors/year; journal; country; risk of bias | Key results | Summary of findings |
|---|---|---|
| **Africa: Sub-Saharan Africa** | | |
| Dauchy, Ross, 2019; Addiction [33]<br>–Kenya<br>Risk of bias: serious | Initiation own-price elasticity, cigarettes:<br>–all, males: -0.03 (95%CI -0.07, -0.00)<br>• lowest wealth tercile, males: -0.02 (95%CI -0.08, 0.04)<br>• Cessation own-price elasticity, cigarettes:<br>–all, males: 0.03 (95%CI -0.26, 0.32)<br>• lowest wealth tercile, males: 0.16 (95%CI -0.75, 0.43) | No evidence of any statistically or economically significant socioeconomic differences in price responsiveness for cigarette smoking initiation or cessation. |
| Kidane, Hepelwa et al., 2017; Applied Econometrics [32]<br>–Tanzania<br>Risk of bias: critical | Total own-price elasticity, cigarette*:<br>Year: 2008; 2010; 2012<br>–Household total expenditures:<br>• very poor: -0.31; -1.21; -2.39<br>• poor: -0.75; -1.50; -1.97<br>• middle: -1.57; 0.17; -0.99<br>• high: -0.29; -0.81; -0.57<br>* Measures of uncertainty/significance level not reported; participation/consumption own-price elasticities not reported. | It is unclear if there were any statistically or economically significant socioeconomic differences in price responsiveness. |
| Americas: Latin America | | |
| Guindon, Paraje, Chávez, 2018; Econ Inq [38]<br>–Argentina<br>Risk of bias: moderate | Initiation own-price elasticity, cigarettes:<br>–Mother's educational level:<br>• primary or less: -0.21 (95%CI -0.55, 0.12)<br>• secondary or less: -0.55 (95%CI -0.84, -0.26)<br>• more than secondary: -0.77 (95%CI -1.10, -0.44) | No evidence of any statistically significant socioeconomic differences in price responsiveness for cigarette smoking initiation. Results, if anything, suggest that lower-SES individuals may have been less responsive to price. |
| Divino, Ehrl et al., 2021; Tob Control [41]<br>–Brazil<br>Risk of bias: critical | Total own-price elasticity, cigarette:<br>–Income:<br>• quartile 1 (low): -0.47 (95%CI -0.68, -0.23)<br>• quartile 2: -0.49 (95%CI -0.69, -0.25)<br>• quartile 3: -0.52 (95%CI -0.71, -0.28)<br>• quartile 4: -0.55 (95%CI -0.75, -0.31)<br>Own-price participation and consumption elasticities not reported. | No evidence of any statistically or economically significant socioeconomic differences in price responsiveness for cigarette smoking. |
| Guindon, Paraje, Chaloupka, 2019; JAMA Pediatr [39]<br>–Chile<br>Risk of bias: moderate | Initiation own-price elasticity, cigarettes:<br>–Mother's educational level:<br>• primary or less: -0.41 (95%CI -0.48, -0.34)<br>• secondary or less: -0.44 (95%CI -0.49, -0.38)<br>• more than secondary: -0.36 (95%CI -0.42, -0.30)<br>–Type of school:<br>• public -0.41 (95%CI -0.47, -0.35)<br>• subsidized: -0.44 (95%CI -0.50, -0.38)<br>• private: -0.30 (95%CI -0.39, -0.20) | No evidence of any statistically or economically significant socioeconomic differences in price responsiveness for cigarette smoking initiation. |
| Gallego, Otalvaro-Ramirez, Rodriguez-Lesmes, 2021; Tob Control [43]<br>–Colombia<br>Risk of bias: critical | Participation own-price elasticity:<br>–Socioeconomic status:<br>• low: -0.70 (95%CI -1.42, 0.01)<br>• mid: -0.62 (95%CI -1.26, 0.01)<br>• high: -0.72 (95%CI -1.41, -0.03) | No evidence of any statistically or economically significant socioeconomic differences in price responsiveness for cigarette smoking participation.<br>Initiation and cessation also explored; no differences between socioeconomic categories were found. |
| Chávez, 2016; Rev Panam Salud Publica [36]<br>–Ecuador<br>Risk of bias: serious | Total own-price elasticity, cigarettes:<br>–Household total expenditures:<br>• tercile 1 (low): -0.25 (95%CI -58, 58)<br>• tercile 2: -1.14 (95%CI -2.10, -0.18)<br>• tercile 3: -1.25 (95%CI -1.84, -0.66) | No evidence of any statistically or economically significant socioeconomic differences in price responsiveness for total cigarette consumption. |
| Paraje, Araya et al., 2021; Tob Control [44]<br>–El Salvador<br>Risk of bias: serious | Total own-price elasticities, cigarettes:<br>–Household total expenditures:<br>• quintiles 1,2 (low): -1.4 (95%CI -35, 32)<br>• quintile 5: -0.9 (95%CI -1.3, -0.5) | No evidence of any statistically significant socioeconomic difference in price responsiveness; estimates are too imprecise to assess whether difference may have been economically significant. |

*(Continued)*

**Table 2.** (Continued)

| Authors/year; journal; country; risk of bias | Key results | Summary of findings |
|---|---|---|
| Franco-Churruarin, Gonzalez-Rozada, 2021; Report (University of Illinois Chicago) [42] –Mexico Risk of bias: serious | Participation own-price elasticity, cigarettes*: –Wealth: • quartile 1 (low): -0.44 (95%CI -0.60, -0.27) • quartile 2: -0.41 (95%CI -0.56, -0.26) • quartile 3: -0.39 (95%CI -0.54, -0.25) • quartile 4: -0.37 (95%CI -0.51, -0.24) • quartile 1 (low): -0.45 (95% CI -0.60, -0.29) • quartile 2: -0.42 (95% CI -0.56, -0.26) • quartile 3: -0.41 (95% CI -0.55, -0.27) • quartile 4: -0.39 (095% CI -0.52, -0.27) * The first set of estimates were obtained using a measure of prices based on self-reported price paid for the last purchase assigned to smokers; and random regression imputation (stochastic regression imputation) to assign price to non-smokers; the second set of estimated were obtained using mean self-reported price by primary sampling unit assigned to smokers and non-smokers. | No evidence of any statistically or economically significant socioeconomic differences in price responsiveness for cigarette participation. |
| Sáenz de Miera Juárez, Guerrero López et al., 2013; Report (Fundación InterAmericana del Corazón México) [35] –Mexico Risk of bias: serious | Total own-price elasticity, cigarettes: –Household income: • tercile 1 (low): -0.60 (95%CI -0.79, -0.41) • tercile 2: -0.60 (95%CI -0.77, -0.42) • tercile 3: -0.55 (95%CI -0.37, -0.73) Participation own-price elasticity, cigarettes*: –Household income: • tercile 1 (low): -0.20 (P < 0.01) • tercile 2: -0.20 (P < 0.01) • tercile 3: -0.11 (P < 0.05) Consumption own-price elasticity, cigarettes*: –Household income: • tercile 1 (low): -0.40 (P < 0.01) • tercile 2: -0.39 (P < 0.01) • tercile 3: -0.44 (P < 0.01) * measures of uncertainty not reported. | No evidence of any statistically or economically significant socioeconomic differences in price responsiveness for total cigarette consumption. |
| Saenz-de-Miera, Thrasher et al., 2010; Tob Control [34] –Mexico Risk of bias: serious | Average no. of cigarettes smoked per day at baseline and follow-up (% change)*: –Education level: • primary graduate or less: -29% • secondary graduate: -27% • high school graduate or more: -33% –Household income: • low: -27% • mid: -35% • high: -27% Quitting at follow-up (relative risks): –Education level (ref: primary graduate or less) • secondary graduate: 1.3 (95%CI 0.5, 3.2) • high school graduate or more: 1.5 (95%CI 0.6, 3.8) –Monthly household income (ref: low) • mid: 1.03 (95%CI 0.48, 2.2) • high: 0.55 (95%CI 0.2, 1.5) * measures of uncertainty not reported. | No evidence of any statistically or economically significant socioeconomic differences in price responsiveness for cigarette consumption or cessation. |
| de los Ríos C, Medina D, Aguilar, 2020; Working Paper (Instituto de Estudios Peruanos) [40] –Peru Risk of bias: serious | Participation own-price elasticity, cigarettes: –Household total expenditures: • tercile 1 (low): -0.70 (95%CI -0.90, -0.08) • tercile 2: -0.26 (95%CI -0.66, 0.16) • tercile 3: -0.30 (95%CI -0.61, 0.02) Participation own-price elasticity, cigarettes: –Household total expenditures: • tercile 1 (low): -1.02 (95%CI -1.12, -0.93) • tercile 2: -0.87 (95%CI -1.10, -0.63) • tercile 3: -0.56 (95%CI -1.01, -0.10) | Point estimates suggest that lower socioeconomic status households were more responsive to price; differences were large enough to be economically significant. No formal assessment of socioeconomic differences in price responsiveness; uncertainty intervals suggest that differences were likely not statistically significant. |

(*Continued*)

**Table 2.** (Continued)

| Authors/year; journal; country; risk of bias | Key results | Summary of findings |
|---|---|---|
| Gonzalez-Rozada, Ramos-Carbajales, 2016; Rev Panam Salud Publica [37] –Peru Risk of bias: serious | Total own-price elasticity, cigarettes: –Household total expenditures: • tercile 1 (high): -0.81 (95%CI -1.09, -0.53) • tercile 2: -0.57 (95%CI -0.70, -0.44) • tercile 3: -0.75 (95%CI -1.03, -0.47) | No evidence of any statistically significant socioeconomic differences in price responsiveness for total cigarette consumption; estimates suggest a possible u-shaped association between socioeconomic status and price responsiveness. |
| **Asia: Eastern Asia** | | |
| Huang, Zheng et al., 2015; Tob Control [46] –China Risk of bias: serious | Consumption own-price elasticity, cigarettes: –Income: • high: -0.15 (95%CI -0.21, -0.09) • mid: -0.14 (95%CI -0.20, -0.09) • low: -0.11 (95%CI -0.22, 0.00) –Education: • Post-secondary: -0.11 (95%CI -0.18, -0.05) • High school: -0.11 (95%CI -0.16, -0.06) • Less than high school: -0.14 (95%CI -0.26, -0.03) | No evidence of any statistically or economically significant socioeconomic differences in price responsiveness for cigarette smoking. |
| Chen, Xing, 2011; China Economic Review [45] –China Risk of bias: critical | Total own-price elasticity, cigarettes*: Year: 1999–2001 –Household income: • tercile 1 (low): -0.46 • tercile 2: -0.42 • tercile 3: -0.42 Year: 1999; 2000; 2001 –Household income: • tercile 1 (low): -0.70; -0.70; -0.37 • tercile 2: -0.57; -0.61; -0.38 • tercile 3: -0.43; -0.51; -0.32 * measures of uncertainty/significance level not reported. | No evidence of any statistically or economically significant socioeconomic differences in price responsiveness for cigarette smoking. |
| **Asia: South-eastern Asia** | | |
| Cheng, Estrada, 2020; Prev Med [48] –Philippines Risk of bias: critical | Participation own-price elasticity, cigarettes* –Education: • < elementary: -0.11 (P < 0.01) • elementary: -0.08 (P < 0.01) • > elementary, ≤ high school: -0.87 (P < 0.01) • > high school: -1.36 (P < 0.01) –Wealth: • tercile 1 (low): -1.79 (P < 0.01) • tercile 2: -1.40 (P < 0.01) • tercile 3: -0.84 (P < 0.01) Consumption own-price elasticity, cigarettes* –Education: • < elementary: -1.0 (P < 0.1) • elementary: not reported (P > 0.1) • > elementary, ≤ high school: 0.19 (P < 0.05) • > high school: not reported (P > 0.1) –Wealth: • tercile 1 (low): 0.14 (P < 0.05) • tercile 2: 0.16 (P < 0.1) • tercile 3: not reported (P > 0.1) * measures of uncertainty not reported. | It is unclear if there were any statistically or economically significant socioeconomic differences in price responsiveness. |
| Quimbo, Casorla et al., 2012; Report (International Union Against Tuberculosis and Lung Disease) [47] –Philippines Risk of bias: critical | Consumption own-price elasticity, cigarettes* –Household expenditures: • deciles 1–3 (low): -1.09 (P < 0.01) • deciles 4–6: -0.80 (P < 0.01) • deciles 7–9: -0.74 (P < 0.01) • decile 10: -0.52 (P < 0.01) * measures of uncertainty not reported. | Point estimates suggest that lower-SES households were more responsive to price; differences were large enough to be economically meaningful. No formal assessment of socioeconomic differences in price responsiveness. |

*(Continued)*

**Table 2.** (Continued)

| Authors/year; journal; country; risk of bias | Key results | Summary of findings |
|---|---|---|
| Jankhotkaew, Pitayarangsarit et al., 2021; Tob Control [49] –Thailand Risk of bias: critical | Participation own-price elasticity, cigarettes*: –Income: • tertile 1 (low): 0.08 (P < 0.05) • tertile 2: 0.06 (P < 0.05) • tertile 3: 0.02 (P < 0.05) Consumption own-price elasticity, cigarettes*: –Income: • tertile 1 (low): -0.61 (P < 0.05) • tertile 2: -0.57 (P < 0.05) • tertile 3: -0.49 (P < 0.05) Participation own-price elasticity, roll-your-own*: –Income: • tertile 1 (low): -0.15 (P < 0.05) • tertile 2: -0.12 (P < 0.05) • tertile 3: -0.19 (P < 0.05) Consumption own-price elasticity, roll-your-own*: –Income: • tertile 1 (low): -0.25 (P < 0.05) • tertile 2: -0.19 (P < 0.05) • tertile 3: -0.18 (P < 0.05) * measures of uncertainty not reported. | No evidence of any statistically or economically significant socioeconomic differences in price responsiveness for cigarette participation or consumption. Point estimates suggest that lower socioeconomic status individuals were more responsive to price for roll-your-own participation; it is unclear if differences were large enough to be statistically or economically significant. |
| **Asia: Southern Asia** | | |
| Huque, Abdullah et al., 2021; Report (ARK Foundation) [55] –Bangladesh Risk of bias: critical | Participation own-price elasticity, cigarettes: –Wealth: • quintiles 1–3 (low): -0.86 (95%CI -1.19, -0.53) • quintiles 4–5: -0.35 (95%CI -0.60, -0.10) Consumption own-price elasticity, cigarettes: –Wealth quintiles: • quintiles 1–3 (low): -0.04 (95%CI -0.14, 0.06) • quintiles 4–5: -0.04 (95%CI -0.10, 0.02) | Point estimates suggest that lower socioeconomic status individuals were more responsive to price for cigarette participation; differences were large enough to be economically significant. No formal assessment of socioeconomic differences in price responsiveness. Uncertainty intervals suggest that differences were not statistically significant. No evidence of any statistically or economically significant socioeconomic differences in price responsiveness for cigarette consumption. |
| Del Carmen, Fuchs, Genoni, 2018; Working Paper (World Bank) [53] –Bangladesh Risk of bias: critical | Total own-price elasticity, cigarettes* –Total household consumption: • decile 1 (low): -1.36 • decile 2: -1.33 • decile 3: -1.33 • decile 4: -1.29 • decile 5: -1.33 • decile 6: -1.27 • decile 7: -1.24 • decile 8: -1.25 • decile 9: -1.25 • decile 10: -1.23 Total own-price elasticity, bidis* –Total household consumption: • decile 1 (low): -1.14 • decile 2: -1.26 • decile 3: -1.18 • decile 4: -1.26 • decile 5: -1.19 • decile 6: -1.18 • decile 7: -1.21 • decile 8: -1.18 • decile 9: -1.27 • decile 10: -1.29 * measures of uncertainty/significance level not reported. | No evidence of any statistically or economically significant socioeconomic differences in price responsiveness for cigarette or bidi smoking. |

*(Continued)*

**Table 2.** (Continued)

| Authors/year; journal; country; risk of bias | Key results | Summary of findings |
|---|---|---|
| Nargis, Ruthbah, et al., 2014; Tob control [51]<br>–Bangladesh<br>Risk of bias: critical | Participation own-price elasticity, cigarettes*:<br>–Socioeconomic status:<br>–Probit:<br>• low: 0.01 (P ≥ 0.05)<br>• mid: 0.00 (P ≥ 0.05)<br>• high: 0.13 (P ≥ 0.05)<br>–IV Probit:<br>• low: -0.50 (P < 0.01)<br>• mid: -0.31 (P < 0.01)<br>• high: -0.15 (P ≥ 0.05)<br>Consumption own-price elasticity, cigarettes*:<br>–Socioeconomic status:<br>–OLS:<br>• low: -0.43 (P < 0.01)<br>• mid: -0.07 (P ≥ 0.05)<br>• high: -0.14 (p ≥ 0.05)<br>– 2SLS:<br>• low: -0.25 (P < 0.001)<br>• mid: -0.09 (P ≥ 0.05)<br>• high: -0.21 (P < 0.01)<br>* measures of uncertainty not reported. | It is unclear if there were any statistically or economically significant socioeconomic differences in price responsiveness. |
| Dauchy, John, 2022; Prev Sci [57]<br>–India<br>Risk of bias: critical | Initiation own-price elasticity, cigarettes/bidis:<br>–Wealth:<br>• tercile 1 (low): -0.025 (95%CI -0.026, -0.025)<br>• tercile 2: -0.025 (95%CI -0.026, -0.025)<br>• tercile 3: -0.018 (95%CI -0.019, -0.017)<br>Chi-squared test of statistical significance between subgroups: low vs middle, P = 0.37; middle vs high, P < 0.01; low vs high, P < 0.01.<br>Initiation own-price elasticity, smokeless tobacco:<br>–Wealth:<br>• tercile 1 (low): -0.0004 (95%CI 0.000, 0.000)<br>• tercile 2: -0.0005 (95%CI 0.000, 0.000)<br>• tercile 3: -0.0004 (95%CI 0.000, 0.000)<br>Chi-squared tests of statistical significance between subgroups: P < 0.01 for all comparisons.<br>Cessation own-price elasticity, cigarettes/bidis:<br>–Wealth:<br>• tercile 1 (low): 0.022 (95%CI 0.014, 0.029)<br>• tercile 2: 0.023 (95%CI 0.016, 0.030)<br>• tercile 3: 0.0211 (95% CI 0.012, 0.030)<br>Chi-squared test of statistical significance between subgroups: low vs middle, P = 0.64; middle vs high, P = 0.89; low vs high, P = 0.60.<br>Cessation own-price elasticity, smokeless tobacco:<br>–Wealth:<br>• tercile 1 (low): 0.003 (95%CI 0.000, 0.005)<br>• tercile 2: 0.004 (95%CI 0.000, 0.007)<br>• tercile 3: 0.001 (95%CI -0.002, 0.004)<br>Chi-squared test of statistical significance between subgroups: low vs middle, P = 0.50; middle vs high, P = 0.42; low vs high, P = 0.70. | Although there were some statistically significant socioeconomic differences in price responsiveness for initiation, differences were very small and unlikely to be economically significant; no evidence of any statistically or economically significant socioeconomic differences in price responsiveness for cessation. |

*(Continued)*

**Table 2.** (Continued)

| Authors/year; journal; country; risk of bias | Key results | Summary of findings |
|---|---|---|
| Selvaraj, Srivastava, Karan, 2015; BMJ Open [52]<br>–India<br>Risk of bias: critical | Total own-price elasticity, cigarettes:<br>–Household total expenditures:<br>• tercile 1 (low): -0.83 (95%CI -0.84, -0.82)<br>• tercile 2: -0.09 (95%CI -0.10, -0.08)<br>• tercile 3: -0.26 (95%CI -0.26, -0.26)<br>Total own-price elasticity, bidis:<br>–Household total expenditures:<br>• tercile 1 (low): -0.43 (95%CI -0.43, -0.43)<br>• tercile 2: -0.25 (95%CI -0.25, -0.25)<br>• tercile 3: -0.08 (95%CI -0.09, -0.07)<br>Total own-price elasticity, leaf tobacco:<br>–Household total expenditures:<br>• tercile 1 (low): -0.56 (95%CI -0.56, -0.56)<br>• tercile 2: -0.45 (95%CI -0.45, -0.45)<br>• tercile 3: -0.05 (95%CI -0.06, -0.04) | Point estimates suggest that lower socioeconomic status households were more responsive to price; differences were large enough to be economically significant.<br>No formal assessment of socioeconomic differences in price responsiveness. |
| Guindon, Nandi, et al., 2011; Working Paper (NBER) [50]<br>–India<br>Risk of bias: serious | Unit values averaged by cluster over all households:<br>Own-price elasticities, bidis:<br>–Household expenditures:<br>• low: -0.95 (95%CI -0.99, -0.91)<br>• high: -0.86 (95%CI -0.94, -0.79)<br>–Education:<br>• ≤ primary: -0.92 (95%CI -0.96, -0.88)<br>• > primary: -0.94 (95%CI -0.99, -0.91)<br>Own-price elasticities, cigarettes:<br>–Household expenditures:<br>• low: -1.11 (95%CI -1.21, -1.02)<br>• high: -0.99 (95%CI -1.05, -0.93)<br>–Education:<br>• ≤ primary: -1.16 (95%CI -1.26, -1.06)<br>• > primary: -0.95 (95%CI -1.03, -0.87)<br>–Unit values averaged by cluster only over households under examination:<br>Own-price elasticities, bidis:<br>–Household total expenditures:<br>• low: -0.95 (95%CI -1.01, -0.90)<br>• high: -0.89 (95%CI -0.95, -0.83)<br>–Education:<br>• ≤ primary: -0.91 (95%CI -0.97, -0.85)<br>• > primary: -0.93 (95%CI -0.99, -0.87)<br>Own-price elasticities, cigarettes:<br>–Household total expenditures:<br>• low: -0.96 (95%CI -1.04, -0.88)<br>• high: -1.02 (95%CI -1.10, -0.94)<br>–Education:<br>• ≤ primary: -1.02 (95%CI -1.09, -0.94)<br>• > primary: -1.00 (95%CI -1.18, -1.02) | No evidence of any statistically or economically significant socioeconomic differences in price responsiveness for cigarette or bidi smoking. |
| Raei, Emamgholipour et al., 2021; Health Econ Rev [56]<br>–Iran<br>Risk of bias: critical | Participation own-price elasticity, cigarettes*:<br>–Household total expenditures:<br>• quintile 1 (low): -0.07 (P < 0.01)<br>• quintile 2: -0.11 (P < 0.01)<br>• quintile 3: -0.12 (P < 0.01)<br>• quintile 4: -0.12 (P < 0.01)<br>• quintile 5: -0.11 (P < 0.01)<br>Consumption own-price elasticity, cigarettes*:<br>–Household total expenditures:<br>• quintile 1 (low): -0.40 (P < 0.01)<br>• quintile 2: -0.36 (P < 0.01)<br>• quintile 3: -0.36 (P < 0.01)<br>• quintile 4: -0.37 (P < 0.01)<br>• quintile 5: -0.32 (P < 0.01)<br>* measures of uncertainty not reported | No evidence of any statistically or economically significant socioeconomic differences in price responsiveness for cigarette smoking. |

(*Continued*)

**Table 2.** (Continued)

| Authors/year; journal; country; risk of bias | Key results | Summary of findings |
|---|---|---|
| Nayab, Nasir et al., 2020; Tob Control [64] –Pakistan Risk of bias: serious | Participation own-price elasticity, cigarettes: –Household total expenditures: • quintiles 1–3 (low): -1.14 (95%CI -1.35, -0.92) • quintile 4–5: 0.10 (95%CI -52, 52) Participation own-price elasticity, chewed tobacco: –Household total expenditures: • quintiles 1–3 (low): -0.75 (95%CI -1.08, -0.41) • quintile 4–5: 0.44 (95%CI -2.7, 3.6) | Unclear if there were any statistically or economically significant socioeconomic differences in price responsiveness for cigarette smoking or chewing tobacco; estimates for higher-SES too imprecisely estimated. |
| **Europe: Eastern Europe** | | |
| Gjika, Zhllima, Imami, 2019; Report (University of Illinois Chicago) [58] –Albania Risk of bias: critical | Participation own-price elasticity, cigarettes: –Household total expenditures: • low: -0.92 (95%CI -1.40, -0.44) • mid: -0.23 (95%CI -0.52, 0.06) • high: -0.35 (95%CI -0.69, -0.02) Consumption own-price elasticity, cigarettes*: –Household total expenditures: • low: -0.28 (95%CI -0.55, -0.01) • mid: -0.15 (95% CI -0.39, 0.09) • high: -0.36 (95%CI -0.62, -0.10) * results obtained using generalized linear models. Results using Deaton's approach not presented. | Point estimates suggest that lower socioeconomic status households were more responsive to price for smoking participation; differences were large enough to be economically significant but may not be statistically significant. No evidence that lower socioeconomic status households were more responsive to price for consumption. |
| Gligorić, Kulovac et al., 2022; Tob Control [63] –Bosnia and Herzegovina Risk of bias: moderate | Participation own-price elasticity, cigarettes: –Household total expenditures: • low: -0.80 (95%CI -0.88, -0.71) • mid: -0.54 (95%CI -0.62, -0.46) • high: -0.33 (95%CI -0.41, -0.25) Chi-squared tests of statistical significance between subgroups: P < 0.01 for all comparisons. Consumption own-price elasticity, cigarettes: –Household total expenditures: • low: -0.61 (95%CI -0.70, -0.51) • mid: -0.45 (95%CI -0.55, -0.36) • high: -0.37 (95%CI -0.49, -0.24) Chi-squared test of statistical significance between subgroups: low vs middle, P = 0.01; middle vs high, P = 0.26; low vs high, P < 0.01. | Point estimates suggest that lower socioeconomic status households were more responsive to price; differences were large enough to be economically significant. |
| Prekazi, Pula, 2019; Report (University of Illinois Chicago) [60] –Kosovo Risk of bias: critical | Participation own-price elasticity, cigarettes: –Household total expenditures: • low: -0.66 (95%CI -2.44, 1.12) • mid: 0.00 (95%CI -2.10, 2.11) • high: -0.47 (95%CI -2.88, 1.95) Consumption own-price elasticity, cigarettes*: –Household total expenditures: • low: -0.53 (95%CI -0.95, -0.11) • mid: -0.63 (95%CI -1.13, -0.13) • high: -0.29 (95%CI -1.25, 0.66) * results obtained using generalized linear models. Results using Deaton's approach not presented. | No evidence of any statistically or economically significant socioeconomic differences in price responsiveness for cigarette smoking. |
| Cizmovic, Mugosa et al., 2022; Tob Control [62] –Montenegro Risk of bias: serious | Participation own-price elasticity, cigarettes: –Household total expenditures: • low: -0.60 (95%CI -0.73, -0.46) • mid: -0.58 (95%CI -0.71, -0.46) • high: -0.34 (95%CI -0.49, -0.20) Chi-squared tests of statistical significance between subgroups: low vs middle, P = 0.77; middle vs high, P < 0.01; low vs high, P < 0.01. Consumption own-price elasticity, cigarettes: –Household total expenditures: • low: -0.42 (95% CI -0.52, -0.33) • mid: -0.34 (95%CI -0.47, -0.21) • high: -0.26 (95%CI -0.54, 0.02) Chi-squared tests of statistical significance between subgroups: P > 0.20 for all comparisons. | Point estimates suggest that low and middle socioeconomic status households were more responsive to price; differences were large enough to be economically meaningful. |

*(Continued)*

**Table 2.** (Continued)

| Authors/year; journal; country; risk of bias | Key results | Summary of findings |
|---|---|---|
| Najdova, 2019; Report (University of Illinois Chicago) [59]<br>–North Macedonia<br>Risk of bias: critical | Participation own-price elasticity, cigarettes:<br>–Household total expenditures:<br>• low: -0.45 (95%CI -0.92, 0.03)<br>• mid: -0.50 (95%CI -0.93, -0.06)<br>• high: 0.19 (95% CI -0.17, 0.55)<br>Consumption own-price elasticity, cigarettes:<br>–Household total expenditures:<br>• low: 0.58 (95%CI -0.20, 1.37)<br>• mid: -0.44 (95%CI -1.46, 0.57)<br>• high: -0.28 (95%CI -1.06, 0.50) | Point estimates suggest that higher socioeconomic status households were less responsive to price for smoking participation; it is unclear if consumption own-price elasticities varied by socioeconomic status. |
| Vladisavljević, Đukić et al., 2019; Report (University of Illinois Chicago) [61]<br>–Serbia<br>Risk of bias: serious | Participation own-price elasticity, cigarettes:<br>–Household total expenditures:<br>• low: -0.57 (95%CI -0.71, -0.42)<br>• mid: -0.26 (95%CI -0.40, -0.12)<br>• high: -0.04 (95%CI -0.17, 0.09)<br>Consumption own-price elasticity, cigarettes:<br>–Household total expenditures:<br>• low: -0.51 (95%CI -0.65, -0.38)<br>• mid: -0.37 (95%CI -0.50, -0.24)<br>• high: -0.22 (95%CI -0.30, -0.14) | Point estimates suggest that lower socioeconomic status households were less responsive to price for smoking participation and consumption.<br>No formal assessment of socioeconomic differences in price responsiveness. Uncertainty intervals suggest that differences were statistically significant. Differences were likely large enough to be economically significant. |

Note: IV, instrumental variable; NBER, National Bureau of Economic Research; OLS, Ordinary Least Square. Geographical regions are based on continental regions, which are further subdivided into sub-regions (United Nations Statistics Division, https://unstats.un.org/unsd/methodology/m49/

61]. However, none of the studies formally examined if socioeconomic differences were statistically significantly different.

An additional four studies, one assessed at serious risk and three at critical risk of bias, provided unclear evidence that own-price effects differed between SES groups in Bangladesh [51], Pakistan [54], the Philippines [48], and Tanzania [32].

All other included studies provide no evidence that SES modified the association between cigarette prices and total cigarette consumption or cigarette smoking participation/consumption (seven studies assessed at serious risk of bias using data from Ecuador [36], El Salvador [44], Mexico [35, 42], Peru [40], China [46], and India [50], and seven studies assessed as critical risk of bias conducted in Brazil [41], Colombia [43], China [45], Thailand [49], Bangladesh [53], Iran [56], and Kosovo [60].

Similarly, included studies provide no evidence of any statistically or economically significant socioeconomic differences in price responsiveness for cigarette smoking initiation in four studies assessed at moderate (Argentina [38] and Chile [39]), serious (Kenya [33]) and critical risk of bias (Colombia [43]) or cessation in three studies assessed at serious (Mexico [34] and Kenya [33]) and critical risk of bias (Colombia [43]). Results from Argentina, if anything, suggest that individuals with lower education may have been less responsive to price than those with higher education attainment [38].

With the exception of one study, assessed at critical risk of bias [52], studies that examined tobacco products other than cigarettes such as bidis, roll-your-own, and smokeless tobacco found that SES was not an effect modifier [49, 50, 53, 54, 57].

A list of excluded studies and reasons for exclusion is provided in S4 Appendix. Of note, we excluded several recent World Bank working papers which reported own-price elasticities by income deciles or quintiles. We excluded these studies because too little methodological information was provided to allow us to assess the risk of bias, or because no original own-price elasticity estimates were provided. Additionally, measures of uncertainty and statistical

significance were generally not provided, while unit values were often naively treated as market prices.

## Risk of bias of included studies

Our assessment of risk of bias is summarized in Table 1 (last column). We did not rate any studies at low risk of bias. We rated three studies at moderate, 13 studies at serious, and 16 studies at critical risk of bias.

Nine studies did not provide a clear description of the dependent (outcome) while seven did not do so for the independent (explanatory) variables. Twenty-four studies did not clearly report how missing data and/or outliers were handled, and 16 studies failed to describe if, and if so how, prices or proxy measures of prices were adjusted for inflation. Studies that relied on spatial variation in unit values or self-reported prices often failed to define or report the number of clusters. Thirteen studies did not report having conducted any sensitivity analyses and only five reported having conducted any testing for misspecification. Of the four studies that used duration analyses, two did not clearly report the functional form of duration/time dependency and how prices were matched to retrospective individual-level data and two likely had informative censoring among younger survey respondents. Lastly, the most important limitation was the lack of formal assessment of socioeconomic differences in price responsiveness. Only seven of 32 studies formally assessed statistically whether own-price effects were modified by SES; three used interactions [38, 39, 46] while four used seemingly unrelated regressions to conduct Wald tests of equality of coefficients across equations [33, 57, 62, 63].

## Discussion

### Main findings

As the World Bank in 1999 and IARC in 2011, we found the evidence was not sufficient to conclude that socioeconomic status modified the relationship between prices or taxes and tobacco use in LMICs. It is important to note that we did not find evidence of no effect. Rather, we conclude that the evidence in LMICs was too limited and methodologically weak to make any conclusive statements.

Our review highlights a number of data and methodological limitations in existing studies. First, as mentioned earlier, most studies used unit values (expenditures/quantity consumed) to construct a measure of price that varied in space (with no or little time dimension). The use of unit values relies on the assumption that there are no price variations within each cluster (e.g., village) which requires both geographical proximity of the households and that households be interviewed at approximately the same time [64]. The variation in unit values within each cluster reflects differences in tobacco product quality and measurement errors. Unit values, unlike market prices, are, at least to some extent, affected by the quality of the product (e.g., higher-SES households may purchase higher priced cigarettes so that unit values may be related to income). Consequently, aggregating unit values over large regions can be problematic. Additionally, because unit values are derived from reported spending and quantity purchased, measurement error in quantity and expenditure will introduce measurement error in the unit value [65]. A recent study found that the own-price elasticity of quantity demand was overstated by a factor of four, on average, if the response of quality to price was ignored [66]. In Papua New Guinea, it was found that just one third of the response to price was from changes in the quantity of tobacco products and two thirds was from quality [67]. Given that the response of quality to price may vary between rich and poor tobacco users, results from studies that ignore quality may under or overestimate differences in price responsiveness between rich and poor. Two studies included in our review presented quality elasticities that varied

substantially by SES categories [37, 44]. Moreover, there is evidence the Deaton approach, which takes quality into account, understates its response [66–68]. An additional difficulty is that to take quality into account when examining socioeconomic differences in price responsiveness, one needs to partition the sample into SES groups which reduces the number of households with positive tobacco expenditures in each cluster, as each cluster should contain at least two consuming households for all goods including in the demand system. It was also often unclear whether all households, at cluster-level, faced the same mean unit values, or a mean unit value specific to each household SES category. In the latter case, households in the same cluster may end-up with different unit values.

Second, duration models are well-suited to study the effects of prices or taxes on smoking onset or cessation. However, incorporating the time or duration dependency is crucial as both theory and data suggest that the hazard rates of smoking onset and cessation are not constant with respect to time [69]. Additionally, a key assumption of duration models is noninformative censoring. Censoring occurs when incomplete information is available about the duration time of some individuals (e.g., an individual being interviewed at 16 years old, before she started smoking at 18) [70]. Noninformative censoring requires that the mechanism that caused censoring was not related to the probability of an event occurring (e.g., starting or quitting smoking). Younger individuals at interview are less likely to have initiated smoking and those who have already started, are less likely to have quit. Hence, younger respondents are more likely to be censored, and censoring is likely to be informative for younger respondents.

Third, the standard statistical approach for potential effect modifiers such as SES is a test for interaction. However, interactions cannot be easily incorporated in demand system approaches that correct for quality and measurement error. Additionally, the interpretation of interaction terms in nonlinear models such as logistic or probit regressions is more complex than in linear models [71, 72]. Most included studies partitioned the sample into two or more SES groups but failed to formally assess if differences in price elasticity estimates were statistically significant. To test whether the coefficients are equal across SES groups, a Wald test can be used [73]. However, standard tests of the equality of coefficients may not be valid in models with binary outcomes such as logistic and probit regressions [74, 75]. Moreover, some measures of associations such as odds ratios should not be compared when they were obtained from different samples from different populations [72].

Our review highlighted a number of data and methodological limitations in existing studies which can inform future research. First, more attention ought to be given to the source and extent of price or tax variation. Second, studies relying on budget shares and unit values should pay particular attention to product quality and measurement errors, avoid averaging unit values over large geographical regions, and clearly report the total number of clusters, the number of clusters with positive expenditures, and the average number (per cluster) of households with positive expenditures for each good examined. Third, for studies that use duration analyses, more attention needs to be given to informative censoring, the functional form of time/ duration dependency, how prices are matched to retrospective data, and the assumption that everyone eventually will fail (e.g., start smoking). Lastly, differences in price responsiveness should be formally assessed.

## Limitations

First, it is difficult to predict the direction of the effect of the overall risk of bias on average own-price effects, let alone on differences in own-price effects between SES categories. To the extent that ill-suited data and weak methods likely yielded estimates that were more imprecise, it is likely the case that weaker studies were less likely to find own-price effects estimates that

differed between SES categories. Second, three included studies were coauthored by an author of this review and eight by recent collaborators of an author of this review. As a result, readers are urged not to rely solely and uncritically on the risk of bias assessment we presented. Third, although a number of the studies reviewed did not clearly provide important methodological information, which rendered risk of bias assessment difficult, we did not contact authors of included studies. Lastly, a review protocol was not made publicly available before starting to work on the review. Risk of bias and quality assessment tools such as ROBIS (Risk of Bias in Systematic reviews) and AMSTAR (A MeaSurement Tool to Assess systematic Reviews) recommend that a review protocol that predates the start of the review be made publicly available [76, 77]. Without such availability, it difficult to assess if eligibility and risk of bias criteria were actually stipulated in advance. We are hopeful that the comprehensiveness and transparency of our search and risk of bias assessment (we documented reasons behind all our risk of bias assessments in detail) and that we did not exclude any studies that met our inclusion criteria because of poor quality will alleviate concerns related to the lack of pre-registration.

## Conclusions

A number of modelling studies have examined the distributional effect of a tax increase on tobacco use, averted treatment costs, catastrophic healthcare expenditures, and/or health in LMICs. Recent examples include modelling studies in China [78], Colombia [79], Ethiopia [80], four Indian states (Karnataka, Assam, Uttar Pradesh and Assam) [81], and a modelling study of male smokers in 13 LIMICs (Armenia, Bangladesh, Brazil, China, Colombia, Chile, India, Indonesia, Mexico, Philippines, Thailand, Turkey, Vietnam) [82]. All studies assumed a strong gradient in own-price elasticity in income. The poor were generally assumed to be more responsive to price by a factor of two to five, relative to the wealthy. Studies that conducted sensitivity analyses with weaker or no income gradient found, unexpectedly, that the distributional consequences changed substantially. Although there are theoretical reasons to expect poorer individuals to be more responsive to monetary prices than wealthy ones in LMICs, our review, as earlier reviews, provides little empirical support [5, 9, 11].

The lack of evidence of socioeconomic differences in tobacco price responsiveness suggests that tobacco tax increases may disproportionally affect poorer individuals if they are not more responsive to increases in tobacco prices than richer individuals. This regressivity in SES may be attenuated/accentuated if the poor consume less/more than the wealth. The focus of policies, however, should not overly focus on the ability to pay, but rather on the overall welfare effects, including, at the very least, the effect on health. For example, poorer individuals in LMICs are more likely to use tobacco and hence, their health burden is relatively greater. A tax increase that raises tobacco prices will then most certainly have a progressive health impact. Additionally, given that the demand for tobacco products is inelastic, revenues generated by higher tobacco taxes can be used to support programs (tobacco-related, or not) that disproportionately benefit the poor.

## Supporting information

**S1 Checklist. PRISMA 2020 checklist.**
(PDF)

**S1 Appendix. Studies from low- and-middle-income countries included in the International Agency for Research on Cancer's report on the effectiveness of price and tax policies for tobacco control (IARC, 2011).**
(PDF)

**S2 Appendix. Search strategy.**
(PDF)

**S3 Appendix. Excluded studies.**
(PDF)

**S4 Appendix. Study characteristics and risk of bias assessment.**
(PDF)

## Acknowledgments

We thank Gioia Buckley, Pranipa Ernest, and Kevin Zhao for their research assistance, and Grieve Chelwa and Guillermo Paraje for their comments and discussion.

## Author Contributions

**Conceptualization:** G. Emmanuel Guindon.

**Funding acquisition:** G. Emmanuel Guindon.

**Investigation:** G. Emmanuel Guindon, Umaima Abbas, Riya Trivedi, Sophiya Garasia, Sydney Johnson.

**Methodology:** G. Emmanuel Guindon, Rijo M. John.

**Project administration:** G. Emmanuel Guindon.

**Supervision:** G. Emmanuel Guindon.

**Validation:** G. Emmanuel Guindon, Umaima Abbas, Riya Trivedi.

**Visualization:** Umaima Abbas, Riya Trivedi, Sophiya Garasia.

**Writing – original draft:** G. Emmanuel Guindon, Umaima Abbas.

**Writing – review & editing:** Riya Trivedi, Sophiya Garasia, Sydney Johnson, Rijo M. John.

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
