## [Decision Letter · Decision Letter 0]

15 May 2023

PGPH-D-23-00510

Socioeconomic differences in the impact of prices and taxes on tobacco use in low- and middle-income countries – a systematic review

Dear Dr. Guindon,

Thank you for submitting your manuscript to PLOS Global Public Health. After careful consideration, we feel that it has merit but does not fully meet PLOS Global Public Health’s publication criteria as it currently stands. Therefore, we invite you to submit a revised version of the manuscript that addresses the points raised during the review process.

We look forward to receiving your revised manuscript.

Kind regards,

Jianhong Zhou

Staff Editor

Journal Requirements:

2. We ask that a manuscript source file is provided at Revision. Please upload your manuscript file as a .doc, .docx, .rtf or .tex.

3. Please provide separate figure files in .tif or .eps format only and remove any figures embedded in your manuscript file. Please also ensure all files are under our size limit of 10MB.

Additional Editor Comments (if provided):

Reviewers' comments:

Reviewer's Responses to Questions

**Comments to the Author**

1. Does this manuscript meet PLOS Global Public Health’s publication criteria? Is the manuscript technically sound, and do the data support the conclusions? The manuscript must describe methodologically and ethically rigorous research with conclusions that are appropriately drawn based on the data presented.

Reviewer #1: Yes

Reviewer #2: Yes

2. Has the statistical analysis been performed appropriately and rigorously?

Reviewer #1: Yes

Reviewer #2: Yes

3. Have the authors made all data underlying the findings in their manuscript fully available (please refer to the Data Availability Statement at the start of the manuscript PDF file)?

Reviewer #1: Yes

Reviewer #2: Yes

4. Is the manuscript presented in an intelligible fashion and written in standard English?

Reviewer #1: Yes

Reviewer #2: Yes

5. Review Comments to the Author

Reviewer #1: I have three comments.

1. I find the description of the rationale for the review unsatisfactory. As stated by the authors in the Methods section, "Studies that examine associations between prices, taxes and tobacco use often use methodological approaches that are overlooked in risk of bias assessment tools. ... Due to heterogeneity between studies, a meta-analysis was not possible." I believe the reader would like to see a discussion on the possible explanations for the choice of the methodological approach and the existence of such heterogeneity among the studies. This is related to comment 2.

2. What is the objective of the review? As stated by the authors, "Our objective was to systematically review quantitative studies that examined if socioeconomic status modified the association between prices and taxes and tobacco use in LMICs." Review the quantitative studies for what purpose? Evaluate the final result? The methodological issues? What are the problems in this field? Insufficient data? Insufficient methodological knowledge by the researchers? This is related to comment 3.

3. I would like to see a more informative discussion regarding the implications for both policy and future research on the issue. It is stated in the Conclusions section that "The lack of evidence of socioeconomic differences in tobacco price responsiveness suggests that tobacco tax increases may disproportionally affect poorer individuals if they are not more responsive to increases in tobacco prices than richer individuals."

The paper's contribution is not so clear. The analysis in the review makes me think that it is the data and mainly methodological problems that drive the results in the reviewed studies. So, I would like to see some recommendations on how to conduct future research on the matter as well as some discussion on the real added value of all these published studies!

Reviewer #2: Thank you for the opportunity to review this interesting paper conducting a systematic review of the literature that considers whether low SES tobacco users in LMIC are more price sensitive. I found the study to be interesting and the topic is certainly of broad interest to the tobacco control community given the importance placed on tobacco tax increases. I found the study to have been well conducted and hence I think it could make a real and important contribution to the literature although I would like to see some changes made first. In particular I think the study suffers from a lack of clarity and information at various points, particularly in terms of methodology and the issue of how bias is identified in the studies highlighted. The specific comments below provide details on my thoughts at relevant points in the paper.

Good luck with the revisions!

Specific points

P.2, the abstract. The 2nd to 4th lines need a little bit more nuance, as I had to read the lines a few times to understand the points being made – I initially thought the wording contradicted itself. I would slightly reword to explain how the tax can be regressive. I also think the need for the review could be more strongly articulated, including why only LMIC were considered – there is nothing about what is already known about the issue in high income countries. That might usefully set the scene and hence the knowledge gap to be investigated.

P.4, last paragraph of the introduction. I think there could be a bit more justification of the study. The final paragraph is quite blunt in that it says what the aim of the study is but doesn’t explicitly link that back to the literature review mentioned above and hence a need. Furthermore, that literature review only considers two large studies, firstly from the Word Bank and then the IARC review – I think it would be good to establish clearly that there are currently no academic reviews conducted of work in this area, and also that further country-level studies have been conducted since the IARC report was published and hence there is a need for such a review.

P.4, methodology. Why did your review only start on 1st Jan 2010? To do so basically means you will be ignoring most, if not all of the literature reviewed in the IARC study which could risk bias. I had expected that the review would consider all work in this area, not just the most recent work. As such I think you need to justify your start timing, even if it is just do so because of practical reasons. I see you return to this later on in page 5 but I do think the impact of this needs to be discussed.

P.7, first paragraph. Why was the review protocol not registered? That is often done with such studies. Furthermore, such studies often adopt the PRISMA guidelines for conducting the study but that isn’t mentioned (although I see there is a check-list linked at the end of the proof). What seems to have been done is in line with those guidelines so I think it would help the reader gain some confidence if they could be explicitly referred to in the methodology section.

P.8, final paragraph of section before study findings. When identifying the funding sources I found the wording a touch cumbersome and think it would have been better to say 13 studies identified Bloomberg funding, and nine identified IDRC funding.

P.8/9, findings. You report on the likelihood of bias in the various studies identified as showing results that indicate a possible SES impact. However, you don’t qualify the direction of the possible bias – it could be that the bias leads to either over or under-reporting of relevant results. That is critical as it speaks to the overall likelihood of studies generating findings of an impact that is real and meaningful in LMIC market contexts. Furthermore, the risk of bias doesn’t in fact mean that the studies were biased – as table 1 shows clearly, most of the issues you highlight are a lack of information being reported rather than something done that is concerning in and of itself.

P.9, 4th paragraph. You present a long list of studies that provide no evidence of SES but don’t comment as to whether any or all of these studies were assessed as being possibly biased or not. Given bias can impact a null result too, I think it is important to be explicit in regards to these studies.

P.9, 5th paragraph. You mention studies looking at tobacco products beyond cigarettes. Did these studies consider multiple tobacco products or were they only concentrated on such particular non-tobacco products. If they do the latter, the studies might in themselves be biased as such non cigarette products are very much used by low SES groups and hence there will inevitably be a bias sample.

P.10, 2nd paragraph of Risk of bias section. As with my comment above I didn’t like the way your wording joined together the description of the dependent and independent variables. Better to separate the wording of these two, i.e. Nine studies did not provide a clear description of the dependent variable while seven did not do so for the independent variable.

P.12, limitations. This feels a bit of an add on rather than a full consideration of the limits of the study herein. I would encourage the authors to expand this to include some of the issues I have outlined above. E.g. what does the lack of the study protocol being published in advance actually mean in practice?

P.13, paragraph 2. What you have included is fine but I think you also need to explicitly allow for another possible scenario - the fact that the lack of evidence might simply be because of the methodological challenges of doing studies in this area that don’t suffer from some methodological limitation.

6. PLOS authors have the option to publish the peer review history of their article (what does this mean?). If published, this will include your full peer review and any attached files.

**Do you want your identity to be public for this peer review?** For information about this choice, including consent withdrawal, please see our Privacy Policy.

Reviewer #1: No

Reviewer #2: No

---

## [Decision Letter · Decision Letter 1]

3 Jul 2023

PGPH-D-23-00510R1

Socioeconomic differences in the impact of prices and taxes on tobacco use in low- and middle-income countries – a systematic review

Dear Dr. Guindon,

Thank you for submitting your manuscript to PLOS Global Public Health. After careful consideration, we feel that it has merit but does not fully meet PLOS Global Public Health’s publication criteria as it currently stands. Therefore, we invite you to submit a revised version of the manuscript that addresses the points raised during the review process.

We look forward to receiving your revised manuscript.

Kind regards,

Chandrashekhar T. Sreeramareddy

Academic Editor

Journal Requirements:

2. Please send a completed 'Competing Interests' statement, including any COIs declared by your co-authors. If you have no competing interests to declare, please state "The authors have declared that no competing interests exist". Otherwise please declare all competing interests beginning with twhe statement "I have read the journal's policy and the authors of this manuscript have the following competing interests:"

Additional Editor Comments (if provided):

Reviewers' comments:

Reviewer's Responses to Questions

**Comments to the Author**

1. If the authors have adequately addressed your comments raised in a previous round of review and you feel that this manuscript is now acceptable for publication, you may indicate that here to bypass the “Comments to the Author” section, enter your conflict of interest statement in the “Confidential to Editor” section, and submit your "Accept" recommendation.

Reviewer #1: All comments have been addressed

Reviewer #2: (No Response)

2. Does this manuscript meet PLOS Global Public Health’s publication criteria? Is the manuscript technically sound, and do the data support the conclusions? The manuscript must describe methodologically and ethically rigorous research with conclusions that are appropriately drawn based on the data presented.

Reviewer #1: (No Response)

Reviewer #2: Yes

3. Has the statistical analysis been performed appropriately and rigorously?

Reviewer #1: (No Response)

Reviewer #2: Yes

4. Have the authors made all data underlying the findings in their manuscript fully available (please refer to the Data Availability Statement at the start of the manuscript PDF file)?

Reviewer #1: (No Response)

Reviewer #2: Yes

5. Is the manuscript presented in an intelligible fashion and written in standard English?

Reviewer #1: (No Response)

Reviewer #2: Yes

6. Review Comments to the Author

Reviewer #1: (No Response)

Reviewer #2: I have reviewed the revised article and I was pleased to see that the authors had engaged with the comments from myself and the other reviewer. I feel most points have been adequately addressed, although I don’t think the authors have done anything more than the absolute minimum in regards to any of the comments received. That is unfortunate as I think a more in-depth engagement with the comments received would have enhanced the article. Furthermore, I also found the authors’ response document to be overly brief and somewhat casual. For example, I felt it did not describe or identify the changes made in sufficient detail. I would suggest in future a more fulsome response would be appropriate.

The one exception to the above is that I note that the authors have still not referenced the PRISMA guidelines within the revised text, even though it is still provided as part of the submission. They also did not explicitly say why this wasn’t mentioned in the article text or why they felt it inappropriate to do so. Furthermore, I feel they could and should have more fully engaged with the point about the lack of pre-registration. I accept the points made in the response document but readers of the paper won’t have that to draw upon. The lack of pre-registrations is briefly mentioned as the final imitation point, but I would have liked to see that expanded to include some discussion of that omission. For instance, why not suggest that the lack of pre-registration was not likely to have materially impacted the quality of the study?

7. PLOS authors have the option to publish the peer review history of their article (what does this mean?). If published, this will include your full peer review and any attached files.

**Do you want your identity to be public for this peer review?** For information about this choice, including consent withdrawal, please see our Privacy Policy.

Reviewer #1: No

Reviewer #2: No

---

## [Editor Report · Decision Letter 2]

8 Aug 2023

Socioeconomic differences in the impact of prices and taxes on tobacco use in low- and middle-income countries – a systematic review

PGPH-D-23-00510R2

Dear Dr. Guindon,

We are pleased to inform you that your manuscript 'Socioeconomic differences in the impact of prices and taxes on tobacco use in low- and middle-income countries – a systematic review' has been provisionally accepted for publication in PLOS Global Public Health.

Best regards,

Chandrashekhar T. Sreeramareddy

Academic Editor

The initial reviews were classified as minor revisions and the authors have satisfactorily addressed all the comments on the earlier version. hence the manuscript is suitable for publication.